# Can causal discovery lead to a more robust prediction model for runoff signatures?

Hossein Abbasizadeh[1], Petr Maca[1], Martin Hanel[1], Mads Troldborg[2], and Amir AghaKouchak[3,4,5]

[1]Faculty of Environmental Sciences, Czech University of Life Sciences Prague, Czech Republic
[2]The James Hutton Institute, Invergowrie, Dundee DD2 5DA, Scotland, UK
[3]Department of Civil and Environmental Engineering, University of California, Irvine, CA, USA
[4]United Nations University Institute for Water, Environment and Health, Hamilton, ON, Canada
[5]Department of Earth System Science, University of California, Irvine, CA, USA

**Correspondence:** Hossein Abbasizadeh (abbasizadeh@fzp.czu.cz)

**Abstract.** Runoff signatures characterize a catchment's response and provide insight into the hydrological processes. These signatures are governed by the co-evolution of catchment properties and climate processes, making them useful for understanding and explaining hydrological responses. However, catchment behaviours can vary significantly across different spatial scales, which complicates the identification of key drivers of hydrologic response. This study represents catchments as networks of variables linked by cause-and-effect relationships. We examine whether the direct causes of runoff signatures, representing independent causal mechanisms, can explain these catchment responses across different environments. To achieve this goal, we train the models using the causal parents of the runoff signatures and investigate whether it results in more robust, parsimonious, and physically interpretable predictions compared to models that do not use causal information. We compare predictive models that incorporate causal information derived from the relationships between catchment, climate, and runoff characteristics. Peter and Clark (PC) causal discovery algorithm is applied separately for 11 runoff signatures to derive causal relationships between catchment attributes, climate indices, and the corresponding runoff signature. Three prediction models, including Bayesian Network (BN), Generalized Additive Model (GAM), and Random Forest (RF), are used for predictions. The results indicate that among models, BN, a linear model with a structure based on the causal network, exhibits the smallest decline in accuracy between training and test simulations compared to the other models. Across nearly all environments and runoff signatures, using causal parents enhances robustness and parsimony while maintaining the accuracy of GAMs. While RF achieves the highest overall performance, it also demonstrates the most significant drop in accuracy between the training and test phases. When the sample size for training is small, the accuracy of the causal RF model, which uses causal parents as predictors, is comparable to that of the non-causal RF model, which uses all selected variables as predictors, particularly for low flow duration, high flow duration, low flows, and high flows. This study demonstrates the potential of causal inference techniques for interpreting and enhancing the prediction of catchment responses by effectively representing the interconnected processes within hydrological systems in a cause-and-effect manner.

# 1 Introduction

Hydrological processes result from complex interactions between climate inputs and catchment characteristics (Sivapalan, 2006). These processes manifest in the catchment response at the catchment outlet. Therefore, catchments can be conceptualized as a unit in which the cumulative effect of all interacting processes defines their runoff behaviour, commonly referred to as "runoff signatures." Runoff signatures encapsulate key characteristics of the runoff process in a catchment, including stream flow magnitude, frequency, and timing. These signatures are essential for a wide range of engineering and scientific applications (Blöschl et al., 2013), especially when causal interpretation or assessment is not possible due to insufficient data. McMillan (2020) outlined a wide range of applications for runoff signatures, such as assessing the performance of hydrological models (Clark et al., 2011; Todorovic et al., 2024), selecting appropriate model structures (Hrachowitz et al., 2014; Spieler and Schuetze, 2024) and estimating parameters (Pokhrel et al., 2012; Pizarro and Jorquera, 2024). They are also instrumental in streamflow prediction in ungauged basins (Yadav et al., 2007; Zhang et al., 2014; Matos and Oliveira e Silva, 2024), and in understanding catchment runoff responses at different spatial and temporal scales (Ficchì et al., 2019).

Although all processes in a catchment contribute to its runoff response, each runoff property (or signature) is directly influenced by a distinct set of climatic and catchment-specific characteristics. As an example, Chagas et al. (2024) studied the regional patterns of low flows across 1400 river gauges in Brazil. They showed that catchment characteristics, especially geological properties, have a significantly greater influence on low flows than climate attributes. Guzha et al. (2018) investigated the effects of changes in forest cover on annual mean flow, high flow, and low flow in 37 catchments of different climatic and physiographic properties in East Africa, concluding that not all catchments exhibit a significant response to forest loss. Therefore, it is necessary to identify a set of variables or covariates that are causally associated with a specific runoff signature and can reliably explain it under various environmental conditions. Understanding these variables allows for explaining the signature of interest across environments with different climatic and physiological conditions.

The main drivers of runoff signatures are commonly investigated using classification and regression methods. These techniques are applied to identify the main drivers influencing catchment response and assess their spatial dependencies. Classification criteria often include runoff properties (Ley et al., 2011; Sawicz et al., 2011; Kuentz et al., 2017), climate, and catchment similarities (Olden et al., 2012; Singh et al., 2016; Yang and Olivera, 2023; Ciulla and Varadharajan, 2024). Additionally, machine learning and statistical methods are widely used for the same purpose. For example, Addor et al. (2018) used random forest to predict 15 runoff signatures across 600 catchments in the US. They showed that climatic attributes are among the most influential predictors of the runoff signatures. McMillan et al. (2022) investigated the dominant process by linking climate and catchment attributes to hydrological signatures over large sets of catchments in the US, UK, and Brazil. They found that although some signatures, such as runoff ratio and baseflow index, were among the most robust metrics for characterizing processes, in some cases, the correlation found among variables and signatures in a country may not always generalize to others. They noted that these diverging correlations could result from statistical associations rather than true causal relationships.

We postulate that investigating the relationship between hydrological variables from causal-and-effect perspectives might solve the problem of diverging correlations reported by McMillan et al. (2022). A variable $X$ is considered the cause of a

variable $Y$ if the value of $Y$ depends on or is influenced by $X$ in any given circumstances (Pearl et al., 2016; Pearl, 2009). Therefore, the probability of a target variable, such as a runoff signature, given its causes, should be the same under different conditions or across different environments. Broadly, there are two widely used frameworks for discovering causal relationships and estimating causal effects from observational data, including structural causal modelling (Pearl, 2009) and the potential

outcome framework (Rubin, 1974). The methods used to discover causality and quantify causal effects and their strength are broadly referred to as Causal Inference Methods (CIMs).

One application of CIMs is to complement machine learning approaches by addressing the problems of transfer and generalization (Schölkopf et al., 2021; Ombadi, 2021), by identifying dependencies and confounding factors using multivariate analyses (Runge et al., 2019a). In an under-investigated cause-and-effect relationship, a confounding variable is an unknown

or unmeasured variable that influences both the supposed effect and the supposed cause (Pearl et al., 2016). Identifying confounders and unravelling causal effects make CIMs a valuable tool for enhancing the interpretability of Earth system models (Reichstein et al., 2019). CIMs are established based on a robust mathematical framework that identifies conditional dependencies in observational data (Pearl, 2009). This process often involves deriving a causal graph based on our understanding of the conditional dependencies among processes using methods such as the Bayesian Network (BN) or Bayesian Belief Network

(Verma and Pearl, 1990).

In the last decade, significant efforts have been made to investigate and develop applications for CIMs in the field of Earth system modelling. These studies, primarily focused on uncovering causal relationships from time series, cover a broad range of topics including climate science (Runge et al., 2019b; Kretschmer et al., 2016), remote sensing (Perez-Suay and Camps-Valls, 2019), soil moisture-precipitation feedback detection (Wang et al., 2018), runoff behaviour (Zazo et al., 2020), the causal

discovery of summer and winter evapotranspiration drivers (Ombadi et al., 2020), and study of hydrological connectivity (Sendrowski and Passalacqua, 2017; Rinderera et al., 2018; Delforge et al., 2022). However, the causal relationships between catchment attributes, climate characteristics, and runoff signatures have yet to be thoroughly investigated. A catchment can be represented as a probabilistic network of interconnected processes leading to a runoff signature. To achieve this, catchments can be conceptualized as Bayesian Networks (BNs), where variables are causally linked. BNs, part of the family of probabilistic

graphical models, consist of nodes representing variables and directed edges indicating causal directions (Koller and Friedman, 2009). The structure of BNs is usually identified through causal discovery methods and expert knowledge (Runge et al., 2019a). Methods for causal discovery, also known as structural learning or causal search, can be categorized into constraint-based, score-based, and asymmetry-based approaches (Runge et al., 2023). Constrained-based methods use conditional independence tests to identify the causal graph, while score-based methods evaluate multiple causal graphs using a scoring function, selecting

the highest-scoring one. Asymmetry-based methods are used to infer causal direction in the bivariate relationships (Runge et al., 2023).

One of the most widely-used causal discovery algorithms is PC, named after its authors Peter Spirtes and Clark Glymour (Spirtes et al., 2001). The PC algorithm is a constraint-based method for causal discovery, which means it infers causal relationships by testing for conditional independencies in the data. It operates under the assumption of causal sufficiency, that is,

all relevant variables are measured and there are no unobserved confounders. Given this assumption and a sufficiently large

sample size, the algorithm guarantees asymptotically correct results Glymour et al. (2019). Although this method is used for discovering Directed Acyclic Graphs (DAGs), its results do not fully identify the true causal structure; instead, it outputs a Markov equivalence class, a set of causal graphs that encode the same conditional independencies. These equivalence classes are typically represented using Completed Partially Directed Acyclic Graphs (CPDAGs) (Peters et al., 2017). Due to its simplicity, computational efficiency, and strong performance, the PC algorithm has been widely adopted across various fields, such as climate science (Ebert-Uphoff and Deng, 2012; Deng and Ebert-Uphoff, 2014), medicine (Sanchez-Romero et al., 2023), and epidemiology Petersen et al. (2021).

The information about the causal relationships between catchment variables can be incorporated into prediction models to predict runoff signatures. Predictions using BNs are primarily designed for discrete datasets that can model complex interactions between variables. The rigorous probabilistic theories involved in BN (Bayesian Network) make them popular for environmental modelling (Aguilera et al., 2011). However, Nojavan et al. (2017) and Qian and Miltner (2015) showed that the results of BNs are influenced by the discretization choice of continuous variables. Inference with BN for continuous variables is still a challenging task (Li and Mahadevan, 2018). Gaussian BN is a widely used method for modelling continuous variables. It assumes that the relationships between variables are linear and variables follow a Gaussian distribution (Marcot and Penman, 2019). To relax these assumptions, non-parametric continuous BNs have been developed (e.g. Qian and Miltner (2015)). However, Gaussian BNs remain a robust and widely-used framework, supported by various software packages (Geiger and Heckerman, 1994). Gaussian BNs have been successfully applied in environmental modelling, particularly for water-quality studies e.g. Jackson-Blake et al. (2022) and Deng et al. (2023).

Given the success of Gaussian BNs in other fields, in this study, we adopt Gaussian BNs to predict runoff signatures. The links between variables of BN are derived from PC causal discovery algorithm. Additionally, two non-linear models, the Generalized Additive Model (GAM) and Random Forest (RF), are used in this study. GAM is an extension of the Generalized Linear Model (GLM) that models non-linear relationships between explanatory and response variables using sums of arbitrary functions of the explanatory variables (Hastie et al., 2009). GAMs have been widely used for hydrological studies including flood frequency analysis (Ouali et al., 2017), low flow frequency analysis (Ouarda et al., 2018), flood peak prediction (Dubos et al., 2022), analysis of nuisance flooding (Vandenberg-Rodes et al., 2016), spatial analysis of extremes (Love et al., 2020), climate-crop yield relationships (Zachariah et al., 2021). RFs, first developed by Breiman (2001), are non-linear, non-parametric models used extensively for regression, classification, prediction, and variable selection. RF-based models have also been used in the field of environmental modelling, including for flow frequency analysis (Desai and Ouarda, 2021), runoff signature prediction (Addor et al., 2018), water level forecasting (Nguyen et al., 2015), downscaling (Arshad et al., 2024), and understanding of drivers of hazards (Seydi et al., 2024).

This study introduces a novel approach for predicting runoff signatures by integrating causal information into predictive models. To the best of our knowledge, causal inference techniques have not yet been applied for this purpose. Unlike previous studies that primarily rely on correlated-based features for predicting a specific catchment response, we take a step beyond mere correlation by focusing on causally relevant variables, specifically, causal parents. By integrating causal information into predictive models (GAM and RF), we aim to investigate whether it can enhance the prediction models' robustness, in-

terpretability, and parsimony compared to models that do not utilize causal insights. We assume that a specific characteristic of catchment response is directly influenced by a subset of correlated variables, known as causal parents, rather than by all correlated variables. These causal parents, together with the runoff signature, form a causal mechanism that is theoretically independent of other variables and can explain the variations in the signature. In this context, our objective is to test whether this fundamental concept applies to complex, real-world hydrological systems. To achieve our objectives, we follow these steps: 1) identify causal relationships between catchment attributes, climate characteristics, and runoff signatures (network structure) using PC causal discovery algorithm (Spirtes et al., 2001), 2) execute models using both the causal parents (causal models) and all selected variables (non-causal models) for entire catchments and subset of catchments, 3) evaluate the robustness of the causal and non-causal models.

## 2 Data

The Catchment Attributes and MEteorology for Large-sample Studies dataset (CAMELS) is used in this study (Newman et al., 2015; Addor et al., 2017). It includes time series of streamflow and hydrometeorological variables, climatic indices (derived from hydrometeorological time series), catchment attributes, and runoff signatures (derived from streamflow time series) for 671 catchments spanning the contiguous United States. The attributes in the CAMELS dataset are divided into 5 categories, including climate, geology, soil, topography, and vegetation (land cover) categories. CAMELS also includes comprehensive explanations of the techniques employed to derive catchment attributes and a discussion of potential limitations in the data sources. The variables used in this study include catchment characteristics, climate attributes, and runoff signatures, which are outlined in Table 1 and Table 2. The streamflow and hydrometeorological time series are not included in this study.

Table 1 includes 41 catchment and climate attributes, including both categorical and continuous variables. In this study, we perform clustering, causal discovery, and prediction, each of which relies on specific assumptions that must be satisfied when applied to our data. Ensuring these assumptions hold is essential for the validity and reliability of the results. All 41 attributes were used in the clustering analysis. However, for the causal discovery and prediction tasks, we used a subset of 22 continuous variables that are most relevant to all runoff signatures. We performed three types of correlation analyses, Pearson, Kendall, and Spearman, to examine the relationships between catchment and climate attributes and runoff signatures. To further assess the predictive power of these attributes, we used the Random Forest algorithm to evaluate feature importance. Variable importance was ranked using the out-of-bag (OOB) approach, based on the Mean Decrease in Accuracy (IncMSE) score. This metric quantifies the increase in prediction error when a variable is excluded from the model, thereby indicating its relative contribution to predicting runoff signatures. The random forest algorithm was implemented using the randomForest package in R (Liaw et al., 2015). The results of this analysis are provided in the Supplementary Materials.

It is important to note that when constructing a causal graph (Directed Acyclic Graph, or DAG) and performing predictions using Bayesian Network methods, which are part of the probabilistic graphical model framework, the selected variables (nodes) must not be deterministic functions of one another, as this would violate the conditional independence assumptions underlying the DAG structure (Koller and Friedman, 2009). For this reason, the aridity index, which is the ratio of precipitation to potential

evapotranspiration, is excluded from the analysis. The issue of determinism also arises when variables are complementary, such as soil texture components, which are represented as percentages of sand, silt, clay, water, organic matter, and other contents that together sum to 100%. To ensure valid causal discovery and prediction, we use only a subset of these variables, specifically sand, silt, and clay, in our analysis.

**Table 1.** Catchment and climate attributes used for clustering, as well as for causal discovery and prediction of all runoff signatures. Variables shown in bold are those selected for causal discovery and prediction. They were calculated over the period from 01/10/1989 to 30/09/2009 (Table 2 in Addor et al. (2018)).

| Category | No | Attribute | Description | Unit |
|---|---|---|---|---|
| Climate | **1** | **p_mean** | Mean daily precipitation | mm/day |
| | **2** | **pet_mean** | Mean daily PET (Priestley-Taylor) | mm/day |
| | **3** | **p_seasonality** | Seasonality and timing of precipitation | - |
| | **4** | **frac_snow** | Fraction of precipitation as snow | - |
| | 5 | aridity | pet_mean/p_mean | - |
| | **6** | **high_prec_freq** | Frequency of high precipitation days | days /year |
| | **7** | **high_prec_dur** | Average duration of high precipitation events (precipitation > 5×p_mean) | days |
| | **8** | **low_prec_freq** | Frequency of high precipitation events (precipitation > 5×p_mean) | days /year |
| | **9** | **low_prec_dur** | Average duration of dry periods (precipitation < 1mm) | days |
| | 10 | high_prec_timing | Season during which most high precipitation days occur (precipitation > 5×p_mean) | season |
| | 11 | low_prec_timing | Season during which most dry days occurs (precipitation < 1mm) | season |
| Topography | 1 | gauge_lat | Gauge latitude | degrees north |
| | 2 | gauge_lon | Gauge longitude | degrees east |
| | **3** | **elev_mean** | Mean elevation of catchment | m |
| | **4** | **slope_mean** | Mean slope of catchment | m/km |
| | **5** | **area_gages2** | Area of catchment | km² |
| Geology | 1 | geol_1st_class | Most common geological class in the catchment | - |
| | 2 | geol_2nd_class | Second most common geological class in the catchment | - |
| | 3 | glim_1st_class_frac | Fraction of most common geological class | - |
| | 4 | glim_2nd_class_frac | Fraction of second most common geological class | - |
| | 5 | carbonate_rocks_frac | Fraction of carbonate rock | - |
| | **6** | **geol_porosity** | Subsurface porosity | - |
| | **7** | **geol_permeability** | Subsurface permeability | m² |

**Table 1.** (continued) Catchment and climate attributes used for clustering, as well as for causal discovery and prediction of all runoff signatures. Variables shown in bold are those selected for causal discovery and prediction. They were calculated over the period from 01/10/1989 to 30/09/2009 (Table 2 in Addor et al. (2018)).

| Category | No | Attribute | Description | Unit |
|---|---|---|---|---|
| Soil | 1 | soil_depth_pelletier | Depth to bedrock (<50m) | m |
| | 2 | soil_depth_statsgo | Soil depth (<1.5m) | m |
| | **3** | **soil_porosity** | Volumetric soil porosity (averaged over the top 1.5m of soil) | - |
| | **4** | **soil_conductivity** | Saturated hydraulic conductivity (harmonic mean over the top 1.5m of soil) | cm/hr |
| | **5** | **max_water_content** | Maximum water content (averaged over the top 1.5m of soil) | m |
| | **6** | **sand_frac** | Sand fraction (averaged over the top 1.5m of soil) | % |
| | **7** | **silt_frac** | Silt fraction (averaged over the top 1.5m of soil) | % |
| | **8** | **clay_frac** | Clay fraction (averaged over the top 1.5m of soil) | % |
| | 9 | water_frac | Fraction of water in 1.5m of topsoil | % |
| | 10 | organic_frac | Fraction of the soil depth marked as organic material (fraction of soil_depth_statsgo) | % |
| | 11 | other_frac | Fraction of other components (fraction of soil_depth_statsgo) | % |
| Vegetation | **1** | **frac_forest** | Forest fraction of catchment | - |
| | **2** | **lai_max** | Maximum monthly leaf area index | - |
| | **3** | **lai_diff** | Difference between maximum and minimum leaf area index | - |
| | 4 | gvf_max | Maximum monthly green vegetation fraction | - |
| | 5 | gvf_diff | Difference between maximum and minimum green vegetation fraction | - |
| | 6 | dom_land_cover | dominant land cover type | - |
| | 7 | dom_land_cover_frac | fraction of dominant land cover | - |

**Table 2.** Runoff Signatures (target variables) in CAMELS dataset, calculated over the period from 01/10/1989 to 30/09/2009

| No | Signature | Description | Unit | Reference |
|---|---|---|---|---|
| 1 | baseflow_index | The ratio of mean daily baseflow to mean daily discharge | - | (Ladson et al., 2013), Table 2 in Addor et al. (2018) |
| 2 | high_q_dur | The average duration of high-flow events (successive days of flow events > 9 × median daily flow) | days | (Clausen and Biggs, 2000), Table 2 in Addor et al. (2018) |
| 3 | high_q_freq | Frequency of high-flow days (flow events > 9 × median daily flow) | days/year | (Clausen and Biggs, 2000), Table 2 in Addor et al. (2018) |
| 4 | low_q_dur | The average duration of low-flow events (successive days of flow events < 0.2 × mean daily discharge (q_mean)) | days | (Olden and Poff, 2003), Table 2 in Addor et al. (2018) |
| 5 | low_q_freq | Frequency of low-flow days (flow events < 0.2 × mean daily discharge (q_mean)) | days/year | (Olden and Poff, 2003), Table 2 in Addor et al. (2018) |
| 6 | q_mean | Mean daily discharge | mm/day | Table 2 in Addor et al. (2018) |
| 7 | Q5 | Low flow: 5% flow quantile (95% exceedance probability) | mm/day | Table 2 in Addor et al. (2018) |
| 8 | Q95 | High flow: 95% flow quantile (5% exceedance probability) | mm/day | Table 2 in Addor et al. (2018) |
| 9 | runoff_ratio | Mean daily discharge to mean daily precipitation | - | (Sawicz et al., 2011), Table 2 in Addor et al. (2018) |
| 10 | slope_FDC | The slope of flow duration curve | - | (Sawicz et al., 2011), Table 2 in Addor et al. (2018) |
| 11 | stream_elast | Steam flow elasticity (sensitivity of annual streamflow to variations in precipitation) | - | (Sankarasubramanian et al., 2001), Table 2 in Addor et al. (2018) |

## 3    Methods

The methodology integrates clustering, causal discovery, and prediction. Fig. 1 shows the methodological procedure used in this study. In Fig. 1, causal models refer to the models that use causal parents, and non-causal models use all 22 variables as predictors (Table 1). Environments are defined as subsets of the dataset obtained through clustering algorithms. Therefore, the word "environment" refers to the clusters or subsets of data. The whole dataset itself is also an environment; however, in this study, we primarily refer to clusters when discussing environments. Baseline models refer to the models that use the whole dataset (i.e., all 671 catchments) for training and testing, and sub-models use subsets of the dataset for this purpose. GAM∼Par and RF∼Par are causal GAM and RF models that employ causal parents for prediction. GAM∼All and RF∼All are non-causal GAM and RF models that use all the selected variables as predictors. A robust model is defined as one that maintains its accuracy across different environments.

In this study, we explore the concept of independent mechanisms in the context of modelling runoff signatures. The independent mechanisms assumption suggests that the causal generative process of a system's variables is made up of self-contained modules that operate independently, without influencing or providing information to one another, and these mechanisms stay stable even when the data distribution changes (Schölkopf et al., 2012; Peters et al., 2017). Using the Directed Acyclic Graph (DAG) obtained from causal discovery, we identified the causal parents of the target runoff signature, which represent the independent causal mechanism generating this variable. Independent mechanisms, as modular components, can be trained separately across different environments and tend to be more adaptable and reusable, a quality we refer to as robustness in this study (Parascandolo et al., 2018). They may also be easier to interpret and provide more insight since these causal mechanisms correspond to physical mechanisms. To evaluate the real-world applicability of this mechanism, we used the identified causal parents as predictors to train RF and GAM. This approach tests whether the independent mechanism derived from the DAG can effectively explain and predict the target variable, supporting the idea that these causal conditionals serve as robust and interpretable modules in the prediction of runoff signatures.

To achieve this, we use the whole dataset for the prediction in baseline models and subsets of the dataset in sub-models, both with and without utilizing causal information, corresponding to causal and non-causal models, respectively. If the causal models performe comparably to or better than non-causal models across different environments, it indicates that causal parents suffice to explain the target variable. In this situation, we can also conclude that the causal discovery is able to recover the direct causes of the runoff signature. In cases where causal models outperform non-causal ones, it suggests that some covariates in the non-causal models may represent spurious correlations, negatively impacting performance in that specific environment. Furthermore, the robustness of the models is assessed by comparing their accuracy in training and test settings and checking whether the difference between causal and non-causal models is statistically significant in both settings. The methods used to calculate statistical significance tests comparing causal and non-causal models are presented in the supplementary material.

The steps are explained in the following sections.

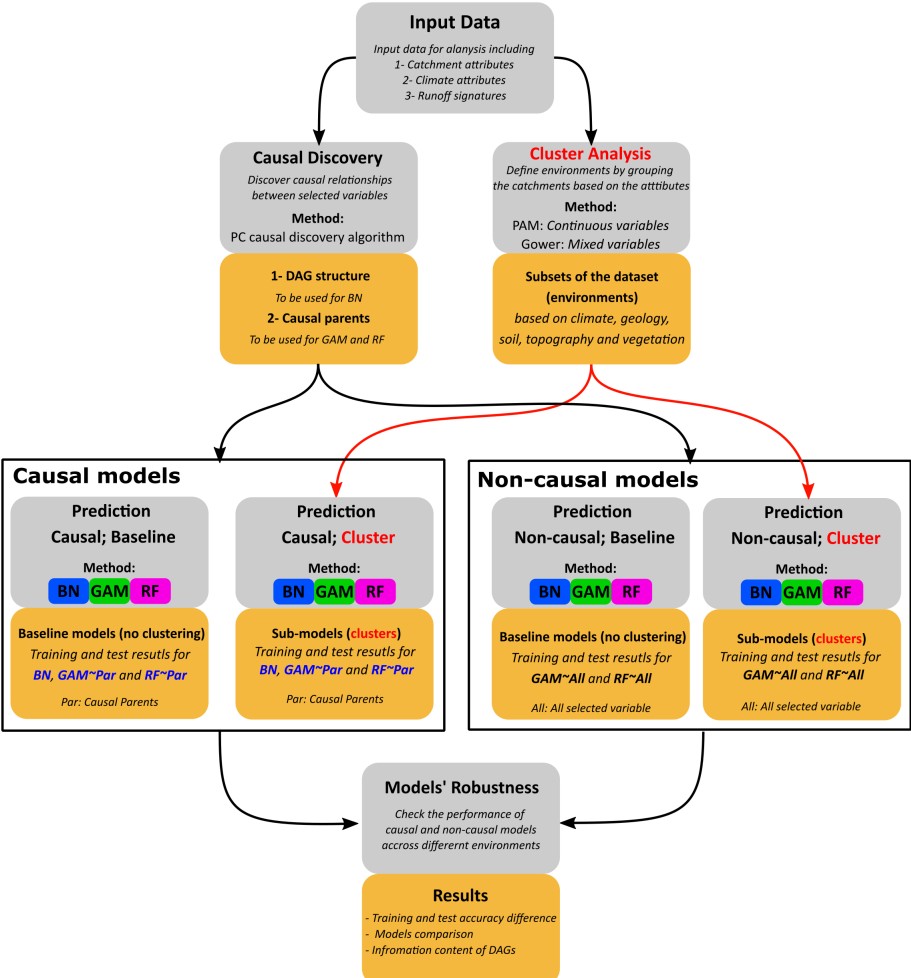

**Figure 1.** Flowchart depicting the steps followed in this study. Grey boxes indicate the procedures, orange boxes present the results of these procedures, blue text highlights where information about causality is utilized, and the red text and arrows highlight the cluster analysis and indicate where the clustering results are applied. PC refers to Peter and Clark's causal discovery algorithm, PAM stands for Partition Around the Centroid clustering algorithm, and DAG refers to Directed Acyclic Graph. BN refers to the Bayesian Network, GAM refers to the Generalized Additive Model, and RF refers to the Random Forest. GAM∼Par and RF∼Par are causal models (GAM and RF) using only causal parent variables for prediction, while GAM∼All and RF∼All are non-causal models that use all selected variables as predictors. Baseline models refer to models that use the entire dataset (all 671 catchments) for training and testing, while sub-models use only subsets of the dataset or clusters.

## 3.1 Clustering

The CAMELS dataset provides five categories of catchment and climate attributes for each catchment (Table 1). Clustering catchments based on each category of attributes is assumed to provide groups of catchments with homogeneous characteristics

(Blöschl et al., 2013). Clustering is used to group the CAMELS catchments into different categories based on specific attributes. Any given catchment will belong to one climate attribute cluster, one soil attribute cluster, one topographic attribute cluster, one geological cluster, and one vegetation cluster (i.e. each catchment is 'assigned' 5 cluster values, one for each attribute). The whole process of training and testing the models is now (also) done on separate attribute clusters only, so basically, it is only done on a subset of the available data, but using data that shares certain characteristics. The causal parents and selected variables are, however, the same whether we use clustering or not.

We investigate the performance of the sub-models within each cluster of catchments. Each cluster is considered a new environment with certain properties to investigate the robustness of models with and without causal parents. The selected covariates remain the same across all environments for each runoff signature. Within each cluster or environment, covariate properties are assumed to be homogeneous with respect to specific attributes, allowing us to train and test models using variables with consistent properties. Defining environments as subsets of data is inspired by Peters et al. (2016). Here, we use clustering analysis to define these subsets, resulting in environments with specific properties. Therefore, clusters can be considered as subsets of data where the distribution of covariates shifts from one cluster to another. This variation across clusters provides a framework for exploring the underlying independent causal mechanisms of each runoff signature.

The causal independent mechanism (the target variable and its parents) for each signature remains unchanged if there is a change in the distribution of parents (Woodward, 2008). Therefore, causal models (models with causal parents as explanatory variables) are expected to perform with consistent accuracy across different environments. This concept is influenced by the covariate shift assumption (Quionero-Candela et al., 2009). Covariate shift states that if variable $Y$ is to be predicted from a set of variables $X$, and $X$ is the cause of $Y$, the properties of conditional probability $P(Y|X)$ remains unchanged the same across all environments if the distribution of $X$ changes. This information will help investigate the performance of the causal compared to the non-causal models.

Two clustering methods are employed to group the catchment attributes in the CAMELS dataset. The K-medoids or Partitioning Around Mediods (PAM) clustering algorithm (Rdusseeun and Kaufman, 1987) is used for categories of attributes with continuous variables, namely, soil and topography. PAM is a more robust method for handling outliers and noise than the K-means method. The Gower distance (Gower, 1971) is used for mixed variables. This method is developed for datasets containing continuous, binary or multiattribute variables (Hennig and Liao, 2013). The elbow and silhouette methods are used to find the optimum number of clusters.

## 3.2 Causal discovery

Causal discovery is used to partially or fully infer the causal structure from observational data or distribution under certain assumptions (Heinze-Deml et al., 2018a). Here, we try to find causal structures from the observational data without specifying the underlying physical equations using a causal discovery method. Causal discovery is applied to each runoff signature along with 22 variables from the CAMELS dataset to identify the causal graph associated with each runoff signature.

### 3.2.1 PC causal discovery algorithm

In this study, the constraint-based PC causal discovery algorithm (Spirtes et al., 2001), named after its authors Peter Spirtes and Clark Glymour is used. The PC algorithm recovers the causal graph from observed data by testing which variables are conditionally independent of each other. These independencies are seen as constraints to satisfy the true data-generating process. The PC algorithm outputs a Completed Partially Directed Acyclic Graph (CPDAG), which represents a Markov equivalence class of causal structures. This means that instead of identifying a single, unique Directed Acyclic Graph (DAG) that fully describes the causal relationships among variables, the PC algorithm returns a set of DAGs that are indistinguishable based on observed statistical dependencies alone. These DAGs all share the same conditional independence, that is, the probability distribution over the variables satisfies the Markov property with respect to each of them. The CPDAG captures this equivalence class by showing which edges are directed and which are only known to exist but not in which direction (undirected edges).

To recover a valid CPDAG using the PC algorithm, certain assumptions must be satisfied to ensure the causal structure is identifiable from the data. These include the causal Markov condition, which links the graph structure to statistical independencies, the faithfulness condition, which ensures that all observed independencies are due to the graph structure or the distribution is faithful to DAG (Peters et al., 2017), and causal sufficiency, which assumes there are no unmeasured confounders.

To construct the causal graph, PC begins with the assumption that every variable is potentially connected to every other variable, forming a fully connected undirected graph. Then, the algorithm starts removing edges between variables based on statistical tests of conditional independence. The key idea is that if two variables are conditionally independent given some set of other variables, then there is no direct causal link between them, so the edge connecting them can be removed. By checking different sets of conditioning variables and removing edges accordingly, the PC algorithm recovers the underlying skeleton of the graph, that is, the undirected structure that shows which variables might have a direct relationship.

Once the skeleton is established, the PC algorithm proceeds to orient edges using specific rules based on the results of the conditional independence tests. A key orientation rule involves identifying v-structures or unshielded colliders. If variables A and B are both connected to a third variable C (i.e., A–C–B), and A and B are not connected to each other, and C was not in the conditioning set that rendered A and B independent, then the structure is oriented as a collider (A $\longrightarrow$ C $\longleftarrow$ B). After identifying all the v-structures, the PC algorithm applies a set of logical rules to orient the edges throughout the graph while avoiding cycles and introducing new v-structures. However, some edges remain undirected when the available conditional independence information is insufficient to determine a unique orientation without introducing ambiguity or inconsistency with the test results. These undirected edges represent causal relationships whose direction cannot be resolved from the data alone without further assumptions (Meek, 1995; Spirtes et al., 2001).

The results of the PC causal discovery algorithm depend on the alpha value (also called the significance level or threshold). The alpha value is used during conditional independence tests to decide if an edge should be removed. If the p-value of a test exceeds the alpha threshold, the algorithm considers the variables to be conditionally independent and removes the edge. Therefore, the value of alpha directly affects the sparsity and structure of the resulting causal graph. The variation in alpha value affects the skeleton and orientation of the graph produced by the PC algorithm (Kalisch et al., 2012). Furthermore, the

sample size affects the results of the PC, as the algorithm relies on statistical tests of conditional independence to determine the graph structure. As the sample size increases, the accuracy of these tests improves, making it more likely for the algorithm to recover the correct causal structure, or more precisely, the correct Markov equivalence class. Identifying the correct structure of a Bayesian network, which relies on independence testing similar to that derived from the PC algorithm, requires substantially more data than approximating the underlying distribution. Underfitting, where true edges are missed, is especially likely when sample sizes are small, as shown in Zuk et al. (2012). Thus, Considering these issues, we aim to address the choice of alpha values and sample sizes in the following sections..

### 3.2.2 Background knowledge and edge assumptions

Background knowledge can significantly enhance causal discovery by reducing ambiguity in the orientation of edges and narrowing the space of plausible causal graphs. According to Perković et al. (2017), incorporating background knowledge, such as known edge directions or variable orderings, into a CPDAG results in a CPDAG which they refer to as maximally oriented partially directed acyclic graphs (maximal PDAGs). Maximal PDAG often contains fewer undirected edges and thus represents a smaller Markov equivalence class. This improves the identifiability of causal relationships and allows for more accurate estimation of causal effects. Similarly, Bang and Didelez (2025) demonstrate that tiered background knowledge, derived from temporal or hierarchical structures, can be encoded as forbidden edge directions to guide constraint-based algorithms. Their algorithms show that such knowledge not only improves the clarity but also reduces the number of independence tests and possible equivalence classes, enhancing both computational efficiency and discovery accuracy. To obtain maximal PDAG and avoid implausible edge orientations, we define some edge assumptions by blacklisting the implausible edges before running PC algorithm. The edge assumptions are as follows:

- Climate variables cannot cause topography variables.

- Soil variables cannot cause climate variables.

- Geological variables cannot cause climate variables.

- Climate variables cannot cause geological variables.

- Vegetation variables cannot cause climate variables, except for potential evapotranspiration.

- Vegetation variables cannot cause topographic variables.

- Vegetation variables cannot cause geological variables.

- Soil variables cannot cause topographic variables.

- Runoff signatures cannot have child nodes.

It is important to note that many of these causally implausible links, such as climate influencing topography, may become plausible when considered over geological timescales. The final assumption, which treats the runoff signature (the target variable) as a sink node by preventing it from causing other variables, implies that its causal parents correspond to its Markov and stable blankets. The Markov blanket of a node consists of its parents, its children, and the parents of its children. Conditioning on the Markov blanket of a node makes the node independent of the rest of the DAG (Pearl, 1988). This setting also makes the target variable's causal parents equivalent to its stable blanket for regression. This is because the causal parents form a subset of the Markov blanket, and interventions on non-parent nodes do not affect the functional relationships underlying the causal mechanism of the target variable (Pfister et al., 2021).

### 3.2.3   Implementation and evaluation of the PC algorithm's results

PC algorithm assumes that the variables are normally distributed. Therefore, the Box-Cox transformation is applied to the data (Dutta and Maity, 2020). Since the results of the PC algorithm can depend on the order of the input variables, we use the PC-stable variant (Colombo and Maathuis, 2014), which addresses this issue by ensuring order-invariant outputs. The bnlearn R package (Scutari, 2009) is used to apply the PC algorithm. Mutual information with the James-Stein estimator (Hausser and Strimmer, 2009) is chosen as the conditional independence test.

In this study, we run the PC algorithm for 22 variables along with a runoff signature. We do this for 11 runoff signatures. The sample size for each variable is roughly 670 data points. Since the result of the PC algorithm is sensitive to the sample size and significance threshold (alpha value), we set the significance threshold to 0.2 to allow for a more inclusive initial edge selection. The high significance level helps reduce the risk of Type I error, that is, missing true causal edges; however, it increases the risk of Type II errors, which is the appearance of false positive edges. The edge assumptions, defined as a blacklist, help prevent the occurrence of a large number of false positive edges. We evaluate the resulting CPDAG following the approach proposed by Petersen et al. (2021). First, we assess the stability of the discovered edges by performing 1,000 bootstrap resamples (Scutari and Nagarajan, 2013) of the data and applying the PC algorithm to each resample, using a stricter significance threshold of 0.05 for conditional independence tests. We then measure the strength of each edge based on its frequency across the bootstrap iterations. The resulting edge strength estimates, which represent the proportion of bootstrap samples in which each edge appears, are then mapped onto the initial CPDAG obtained from the PC algorithm. This approach enabled us to evaluate the stability and sparsity level of the causal links if we had a dataset with a larger sample size. Then, we use regression modelling with cubic splines as a heuristic test for conditional independence, assessing the statistical significance of each edge present in the CPDAG produced by the PC algorithm. We fit cubic spline regression models for each pair of variables $(X_i, X_j)$, using a conditioning set $\mathbf{Z} = \{X_1, \ldots, X_m\} \, \epsilon \, \mathbf{X} \setminus \{X_i, X_j\}$. We regress $X_i$ once on $\mathbf{Z}$, and once on $\{X_j\} \cup \mathbf{Z}$ (forward direction). To account for potential asymmetry in the relationships, we repeat the same procedure by regressing $X_j$ on $\mathbf{Z}$ and on $\{X_i\} \cup \mathbf{Z}$ (backwards direction). In total, we fit four models for each pair of adjacent variables in the CPDAG, which are as follows:

$$M_{0a} : g(X_i) = \sum_{k=1}^{m} s_k(Z_k)$$

$$M_{1a} : g(X_i) = s_0(X_j) + \sum_{k=1}^{m} s_k(Z_k)$$

$$M_{0b} : \tilde{g}(X_j) = \sum_{k=1}^{m} \tilde{s}_k(Z_k)$$

$$M_{1b} : \tilde{g}(X_j) = \tilde{s}_0(X_i) + \sum_{k=1}^{m} \tilde{s}_k(Z_k)$$

where $g$ and $\tilde{g}$ are identity functions, $s$ and $\tilde{s}$ are cubic splines. The likelihood ratio tests are applied to compare $M_{0a}$ and $M_{1a}$ as well as $M_{0b}$ and $M_{1b}$ to estimate the conditional independence between pairs of adjacent nodes. We consider variables $(X_i, X_j)$ adjacent if at least one of the following null hypotheses is rejected.

$$H_{0a} : M_{0a} = M_{1a} \quad and \quad H_{0b} : M_{0b} = M_{1b}$$

Since no distributional assumptions are made, this test establishes a necessary, but not sufficient, condition for conditional independence between variables (Petersen et al., 2021). Here, we report the p-values obtained from the likelihood ratio test for each edge in the DAGs, without assuming a specific alpha threshold. The alpha value can be set to 0.05 to be compared with the significance threshold used in bootstrapping DAGs, or to a value between 0.05 and 0.2 to be compared with the edges identified in the initial PC algorithm run. The p-value provides an insight into the trade-off between statistical significance and edge strength.

The DAG for each runoff signature is derived from the corresponding CPDAG by orienting the undirected edges. It is done in a way that avoids introducing new unshielded colliders—structures in which two parent nodes point to a common child without being connected to each other. Introducing such colliders would change the set of conditional independencies encoded by the original CPDAG.

## 3.3 Prediction models

The obtained DAG structures are used to predict runoff signatures using Bayesian Network (BN) methods. Additionally, Generalized Additive Models (GAM) and Random Forests (RF) are applied to predict runoff signatures: once using all variables in the DAGs (non-causal models) and once using only the causal parents of the target nodes (causal models).

### 3.3.1 Bayesian Network (BN)

Having the graph structure from the causal discovery algorithm, the data is fitted to the graph, and the parameters are estimated. Gaussian BN is used for inference purposes. Gaussian BN belongs to the family of continuous BNs, meaning the nodes are continuous variables. The conditional dependencies are linear and follow the joint Gaussian distribution. The prediction is made using averaging likelihood simulation with 500 random sampling numbers. Averaging likelihood simulation is a particle-based approximate method for inference in probabilistic graphical models. This method calculates the weight of samples according to the likelihood of evidence, which is a specific value of the signature of interest. It adds up these weights for each sample

(Koller and Friedman, 2009). Since Gaussian BN is limited to capturing only linear relationships, other non-linear prediction methods are also employed in this study, which are explained in the following sections.

### 3.3.2 Generalized Additive Model (GAM)

The Generalized Additive Model (GAM) model (Hastie et al., 2009) is also chosen to handle non-linear relationships between predictors and runoff signatures. GAMs are extensions of Generalized Linear Models (GLMs), which can identify the linear and nonlinear relationship between response and explanatory variables. This method uses scatterplot smoothers (e.g., smoothing spline or kernel smoother) to fit the additive functions. In this study, the penalized regression spline is used as the smoother. This smoother prevents the model from overfitting where the coefficients of penalized spline decrease (Dubos et al., 2022). The calculation is done using mgcv R package (Wood, 2018). we used cubic regression splines for the smooth terms. The outcome variable is continuous, and we used the default identity link function with a Gaussian error distribution. The GAMs were fitted using Restricted Maximum Likelihood to estimate the smoothing parameters. The model predicts the signatures once with all variables derived from feature selection (non-causal model) and once with only the causal parents of the signatures derived from the causal discovery section (causal model).

### 3.3.3 Random Forest (RF)

The last prediction model used in this study is Random Forest (RF). This method estimates response variables using multiple regression trees. Besides its ability to identify nonlinear patterns in the data, the likelihood of overfitting in RF is low because the model's prediction is an ensemble of multiple predictions. Therefore, it can deliver an accurate prediction with little computational effort. These features in the RF model help identify the issues of linearity and overfitting in BN and GAM models, respectively. The randomForest R package (Breiman, 2018) is used with the number of trees set to 500 to stabilize the prediction (Addor et al., 2018). Similar to GAM, RF is run twice: once using all selected variables as the predictors of the runoff signature (non-causal model) and once using only the causal parents as predictors (causal model).

For all models, BN, GAM, and RF, the environments are divided into training and test sets, where 75% of the catchments are randomly selected for training, and the remaining 25% are used for testing. This process is repeated 100 times using bootstrapping to generate different combinations of training and test sets. This approach provides a range of model performances, and their average performance is used for comparison. Importantly, the training and testing of models are conducted within the same environment, meaning that models trained for a specific environment are tested within that same environment. For example, if a model is trained on catchments from a specific climate category cluster, it is also tested on catchments within that same cluster. The models are executed for the whole dataset (baseline models) and each cluster of categories (sub-models). The models' accuracy is evaluated using Root Mean Squared Error (RMSE) and R-squared metrics between prediction and observations. The iteration provides 100 RMSE and R-squared for each run, and the accuracy is reported as their mean value. The following section discusses the obtained results of this study.

## 4 Results

### 4.1 Clustering results for each category

The clustering classifies the catchments according to the five categories. Time series data is not used for clustering analysis, and only catchment attributes available in the CAMELS dataset, as listed in Table 1, are utilized for this purpose. Table 3 shows the methods used for clustering, the optimum number of clusters according to the elbow and Silhouette scores, and the number of catchments in each cluster. Fig. 2 illustrates each cluster's spatial extent of catchments along with two chosen variables. The obtained results from the cluster analysis for each category of attributes are as follows:

1. **Climate attributes:** Climate attributes in the CAMELS dataset are derived from area-weighted averaging of meteorological forcing time series from October 1, 1989, to September 30, 2009. The cluster analysis shows four distinct climate categories, which spread in the east (cluster 1), the Midwest (cluster 2), the west (cluster 3) and the northwest (cluster 4) (Fig. 2a). The largest group of catchments belong to cluster number one, with 334 members in the north- and southeast of the US (Table 3). This cluster receives an average of 3.5 mm daily precipitation and has 2.8 mm daily evapotranspiration. Other clusters have the following average precipitation and evapotranspiration levels: Cluster 2 has 2.3 mm of precipitation and 2.7 mm of evapotranspiration, Cluster 3 has 5.5 mm of precipitation and 2.4 mm of evapotranspiration, and Cluster 4 has 2.0 mm of precipitation and 3.3 mm of evapotranspiration.

2. **Soil attributes:** The soil properties data, derived from the State Soil Geographic Database (STATSGO), provides information about the top 2.5 meters of soil. However, the CAMELS dataset only includes soil data for the top 1.5 meters. Soil texture is represented in 16 classes, of which there are 12 classes based on the United States Department of Agriculture (USDA) and 4 non-soil classes. The saturated hydraulic conductivity and soil porosity are calculated based on the sand and clay fraction using multiple regression analysis. Cluster analysis identifies six groups of catchments. There is no distinctive spatial pattern among soil clusters. However, clusters 2 and 3 are mostly spread across the east and west coastlines (Fig. 2b). The maximum water content and porosity values are influenced by soil texture, which defines the proportion of sand, clay, silt, and other materials. For example, cluster 6 shows the highest soil porosity and maximum water content (Fig. 2b). This cluster has the highest percentage of clay (26%) and silt (47%) fractions among all clusters.

3. **Topographic attributes:** The topographic information of catchments, namely catchments' contours, is determined using geospatial fabric (Viger and Bock, 2014) and Geospatial Attributes of Gages for Evaluating Streamflow (GAGES II) methods (Falcone, 2011). These methods are used to determine the area, and the Digital Elevation Model (DEM) is clipped for each catchment. This category is divided into 4 distinctive clusters. Cluster 1 contains catchments located in the northeast, which are catchments with low elevation and slope (Fig. 2c). Cluster 2 consists of catchments along the west coast spread from the west to the northwest. The catchments with the lowest elevation and slope are in cluster 3, located in the southeast. Cluster 4 contains the highest elevation catchments in the Rocky Mountains (Fig. 2c).

4. **Geological attributes:** The geological variables in the CAMELS datasets are derived from the Global Lithological Map (GLiM) (Hartmann and Moosdorf, 2012) and the Global HYdrogeology MaPS (GLHYMAPS) (Gleeson et al., 2014). From the GLiM dataset, sixteen lithological classes are identified, and their proportional areas are calculated for each catchment. The GLHYMAPS dataset is used to estimate subsurface permeability and porosity (Addor et al., 2017). This category is divided into 7 groups. Unlike the climate and topography categories, this category does not show a distinguishable spatial pattern (Fig. 2d). However, the catchments with the highest geological porosity are mainly concentrated in the southeast, and those with the lowest are located in the west (Fig. 2d).

5. **Vegetation attributes:** Vegetation is represented using two indicators, vertical density, measured by the Leaf Area Index (LAI), and horizontal density, measured by the Green Vegetation Fraction (GVF). These measurements are derived from a 1-km resolution product of the Moderate Resolution Imaging Spectroradiometer (MODIS). The vegetation or land cover category is divided into 6 different groups (Fig. 2e). The spatial pattern of the vegetation is influenced by climate and topographic categories. According to Fig. 2e, the catchments with the highest forest fractions have the highest maximum leaf area index and are located in the northeast and east of the study area. This area has high precipitation and low evapotranspiration (Fig. 2a). The lowest vegetation cover belongs to the central and southern parts of the US, which are in clusters 4 and 6.

These clusters are subsets of the CAMELS dataset with specific properties and different numbers of catchments to be used for runoff signature prediction. They help evaluate the models' performance in different environments, analyse the effect of causal parents as predictors, and assess how the number of data points impacts the training and test simulations.

**Table 3.** Attribute categories, clustering methods, number of clusters, and catchments per cluster.

| No | Category | Method | No. of cluster | No. of Catchments |
|---|---|---|---|---|
| 1 | Climate | Gower | 4 | 334, 144, 87, 103 |
| 2 | Soil | PAM | 6 | 154, 123, 138, 88, 95, 70 |
| 3 | Topography | PAM | 4 | 282, 119, 117, 90 |
| 4 | Geology | Gower | 7 | 149, 53, 123, 116, 64, 104, 42 |
| 5 | Vegetation | Gower | 6 | 89, 131, 149, 69, 105, 128 |

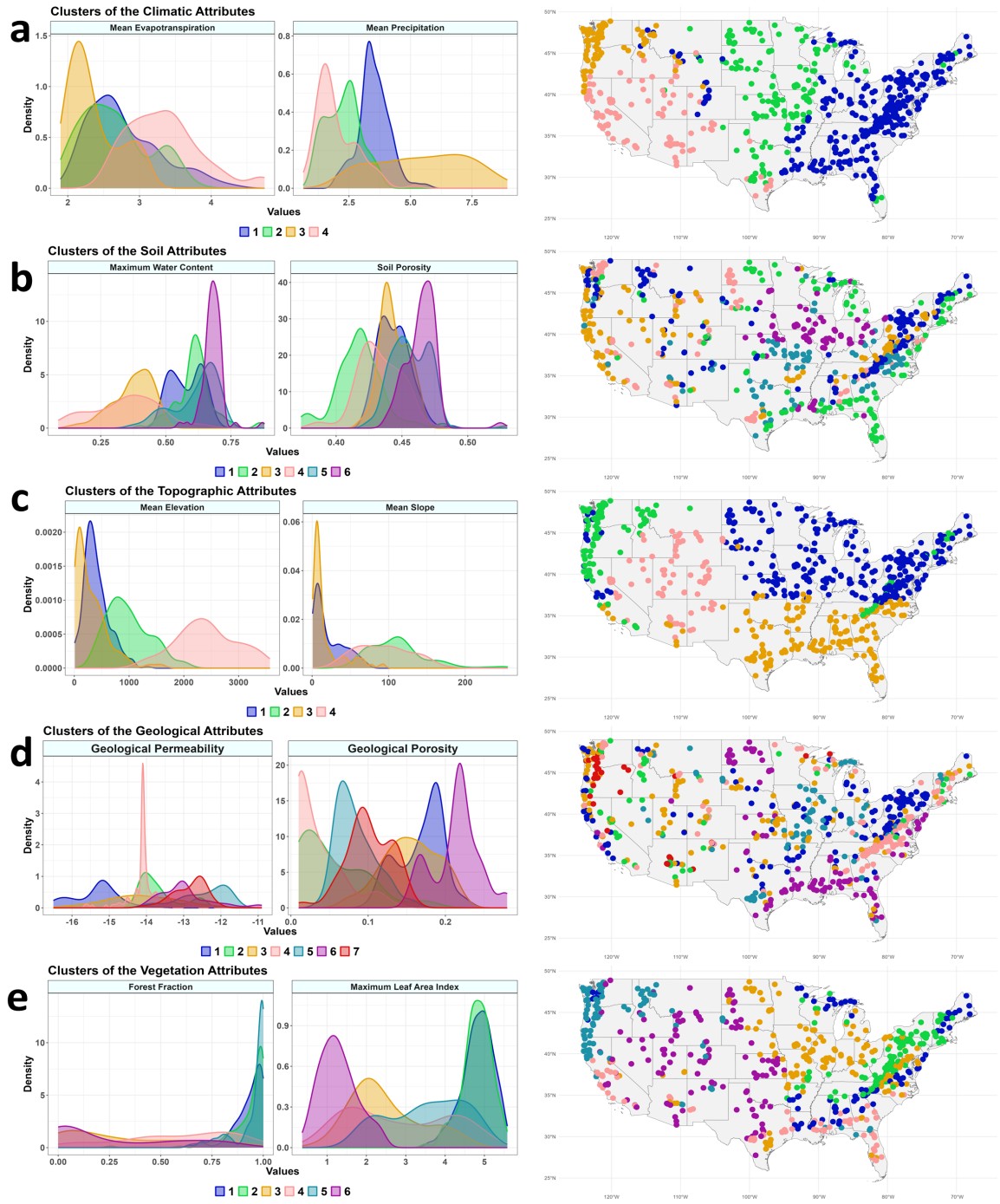

**Figure 2.** The spatial pattern of clusters (right column) and the density of two variables of its corresponding category (left column). The plots show spatial pattern of a) climate attributes, b) soil attributes, c) topographic attributes, d) geological attributes, and 5) vegetation or landcover attributes.

## 4.2 Identification of Causal Links

PC algorithm results identify the causal links between all variables. The output of the PC algorithm is a Completed Partially Directed Acyclic Graph (CPDAG), which may contain undirected edges. In all CPDAGs obtained in this study, the connection between mean elevation and mean slope remains undirected. To derive a fully directed acyclic graph (DAG) from the CPDAG, we orient this edge from mean elevation to mean slope. As shown in Fig. 3b, this orientation does not introduce any unshielded colliders or cycles in the graph. Therefore, we ensure that the resulting DAG belongs to the same Markov equivalence class as the CPDAG produced by the PC algorithm.

Fig. 3a shows the obtained DAG for the baseflow index. The signature (red node) has four direct causes or parents (yellow nodes). Fig. 3c shows the nodes that form the independent causal mechanism for the baseflow index, which are shown by the green line. The identified causal parents of the base flow index include variables related to catchment storage, such as groundwater and snow storage, which are physically meaningful. For each recovered edge, we report both the edge strength and the p-value from the likelihood ratio test, as shown in Fig. 3a. The causal models, ~Par, are trained within the causal mechanism to predict the baseflow index (Fig. 3b). The causal parents in the independent mechanisms also form the Markov and stable blankets for the baseflow index. The structure and variables of the DAG remain unchanged across all environments; only the values of the variables change across environments. DAGs can show the order in which the variables are connected. For instance, the climate and vegetation variables in Fig. 3a are controlled by topographic attributes, which are mean elevation and mean slope. These variables are independent of other categories in the DAG since they do not have any parents delong to the other categories. Furthermore, the causal parents of the signatures, which are identified by the PC algorithm, are not necessarily the most influential variables derived from correlation and variable importance analysis (see Supplementary Materials Section 1).

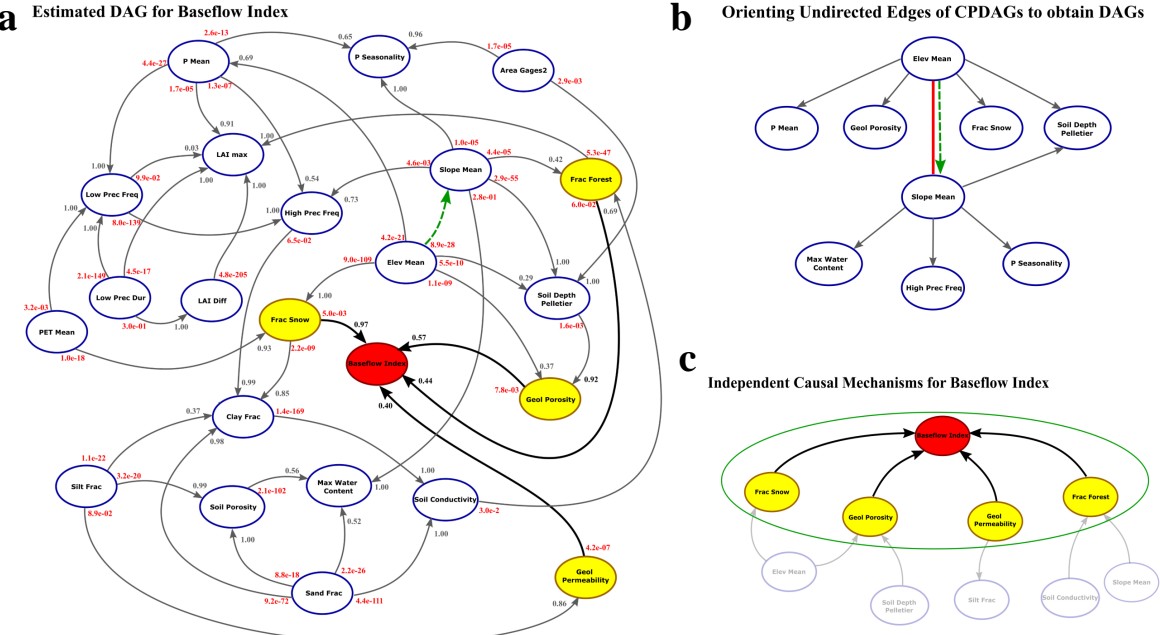

**Figure 3.** a) Directed Acyclic Graph (DAG) for the Baseflow Index. Arrows indicate the causal links between variables. The green dashed arrow represents an oriented edge that was originally undirected in the CPDAG derived from the PC algorithm. The red node denotes the target variable (runoff signature), the yellow nodes represent its causal parents. Red numbers at the beginning of each arrow correspond to the p-values from the likelihood ratio tests, and the grey (or black for the target variable) numbers indicate the edge strengths derived from 1000 bootstrap resamples. The node variables are explained in Table 1. b) The part of the CPDAG that contains an undirected edge. Orienting this edge does not introduce any new unshielded colliders. c) The independent causal mechanism for the baseflow index, which is represented by the red and yellow nodes.

Table 4 shows the causal parents, the p-value of the likelihood ratio test, and the edge strength for each runoff signature.. The number of parents varies from 2 to 5 variables. We compared the performance of the models using only parents (causal models) to the models using all the selected variables as explanatory variables (non-causal models). The models are executed for the 671 catchments as baseline models and for each cluster as sub-models. The results reveal the models' behaviours in different environments (clusters) compared to the baseline models.

**Table 4.** Causal parents of the runoff signatures and their numbers derived from the PC algorithm. P-values are obtained from the likelihood ratio test, and the edge strengths are derived from the frequency of edges that appear in 1000 bootstrap resamples.

| Signature | Detected causal parent by PC (p-value; Edge strength) |
|---|---|
| baseflow_index | geol_permeability ($4.2e-07$; $0.40$), frac_snow ($5.0e-03$, $0.97$), frac_forest ($6.0e-02$, $0.44$), geol_porosity ($7.8e-03$, $0.57$) |
| high_q_dur | p_mean ($3.7e-08$, $0.56$), lai_diff ($5.6e-04$, $0.95$) |
| high_q_freq | low_prec_freq ($2.4e-01$, $0.79$), geol_porosity ($6.8e-02$, $0.52$) |
| low_q_dur | low_prec_dur ($3.0e-08$>, $0.74$), max_water_content ($2.1e-03$, $0.76$), frac_snow ($2.9e-04$, $0.88$) |
| low_q_freq | frac_snow ($2.7e-06$, $0.97$), low_prec_freq ($1.6e-02$, $0.45$), frac_forest ($1.0e0$, $0.71$), geol_porosity ($8.0e-01$, $0.66$), geol_permeability ($2.5e-06$, $0.27$) |
| q_mean | p_mean ($5.3e-160$, $1.0$), p_seasonality ($3.2e-03$, $0.85$), area_gages2 ($4.8e-02$, $1.0$), frac_forest ($6.2e-02$, $0.63$), geol_porosity ($1.0e-06$, $0.79$) |
| Q5 | p_mean ($4.3e-18$, $0.77$), low_prec_freq ($2.7e-10$, $0.67$), slope_mean ($6.8e-02$, $0.82$), geol_porosity ($1.6e-01$, $0.57$), geol_permeability ($3.9e-05$, $0.26$) |
| Q95 | p_mean ($7.6e-96$, $1.00$), p_seasonality ($1.5e-01$, $0.98$), low_prec_freq ($1.4e-03$, $0.81$), area_gages2 ($6.3e-02$, $0.35$), frac_forest ($1.3e-01$, $0.57$) |
| runoff_ratio | p_mean ($1.0e-34$, $0.83$), p_seasonality ($6.8e-05$, $0.97$), area_gages2 ($2.6e-01$, $1.00$), geol_porosity ($4.8e-06$, $0.97$) |
| slope_FDC | p_mean ($3.7e-07$, $0.92$), pet_mean ($4.1e-09$, $0.40$), low_prec_freq ($3.7e-06$, $0.39$) |
| stream_elast | pet_mean ($7.1e-02$, $0.32$), clay_frac ($2.5e-01$, $0.41$), frac_snow ($1.6e-02$, $0.79$), frac_forest ($4.7e-05$, $0.81$), area_gages2 ($1.4e-01$, $0.41$) |

The obtained DAGs, presented in the Supplementary Materials, reveal consistent causal relationships between catchment and climate attributes across all runoff signatures, although the strength and significance of the edges vary. In all cases, topographic variables directly influence climate, vegetation, soil, and geological attributes. Climate variables influence vegetation and all runoff signatures (Table 4). Only two edges connect climate and soil variables, specifically, from high precipitation frequency and the fraction of snow to the clay fraction. Across all DAGs, these edges consistently exhibit high strength and statistical significance. Soil variables influence vegetation, specifically the fraction of forest, as well as geological variables. However, these variables act as causal parents for only two runoff signatures, namely low flow duration and streamflow elasticity. Vegetation attributes do not drive other catchment attributes. They are influenced by climate, topography, and soil variables. However, they directly affect six runoff signatures, including baseflow index, high flow duration, low flow frequency, mean flow, high flow, and streamflow elasticity (Table 4). The geological variables are influenced by topography and soil. They are among the causal parents of baseflow index, high flow frequency, low flow frequency, mean flow, low flow, and runoff ratio (Table 4). The edges

in the obtained DAGs are generally characterized by high strength and statistical significance. However, the link between low precipitation frequency and maximum leaf area index stands out as the weakest, with a strength of less than 1% and a marginal significance level (p-value slightly below 0.1). Notably, this link is absent in the DAGs corresponding to high flow duration, high flow frequency, low flow frequency, and the slope of the flow duration curve.

## 4.3 Performance of the baseline models (prediction using the whole dataset)

The models' performance is evaluated according to the value of RMSE, R squared between observation and prediction, and the differences between the training and test results. The obtained results for each signature are shown in Fig. 4, Table A1, and Fig. 5. The results are derived from the simulation using the whole dataset (671 catchments), which we call the baseline. Baseline models are considered the most accurate models, in which 75% of the whole dataset is used for training and 25% for test simulation. The training and test sets are randomly sampled 500 times, and models are executed after each sampling. The grey dots in Fig. 4 indicate the simulation results for each model's execution. The simulation for GAM and RF models is done twice, once using all the predictors, which are shown by GAM~All and RF~All (non-causal models), and once using only causal parents as predictors, GAM~Par and RF~Par (causal models).

Fig. 4 and Table A1 show that reducing the number of predictors decreases the models' accuracy. Among all models, RF models are the most accurate despite showing the most significant drop in accuracy between training and testing simulations (Fig. 5). The R-squared values from the non-causal RF model (RF~All), in which all selected variables are used as predictors, are compatible with the results obtained from the study of Addor et al. (2018). Using causal parents for RF simulations (RF~Par) leads to a greater distance between training and test results compared to using RF~All for some signatures. These signatures are baseflow index, runoff ratio, and the slope of flow duration curve with 38%, 53%, and 25% increases in distance, respectively, caused by using causal model (Fig. 5). The causal model slightly reduces the gap between training and test results for low-flow duration, low-flow frequency, and high-flow magnitude, with improvements of 7%, 6%, and 1%, respectively. Similar to the RF model, the accuracy of GAM models is decreased by reducing the number of predictors from all selected variables to parent variables (Table 4 and Fig. 5). However, unlike RF, the distance between the training and test accuracy in R squared versus RMSE space significantly decreases by using the causal model for the GAM (Fig. 5). This distance decreases from 41% for the slope flow duration curve to 87% for the high flow frequency (Fig. 5). Finally, BN is the least accurate model in capturing the variance since it is a linear model; however, it shows almost the same R-squared and RMSE values in training and testing simulations. As seen in Fig. 5, BN has the shortest distance between training and test compared to the other two models.

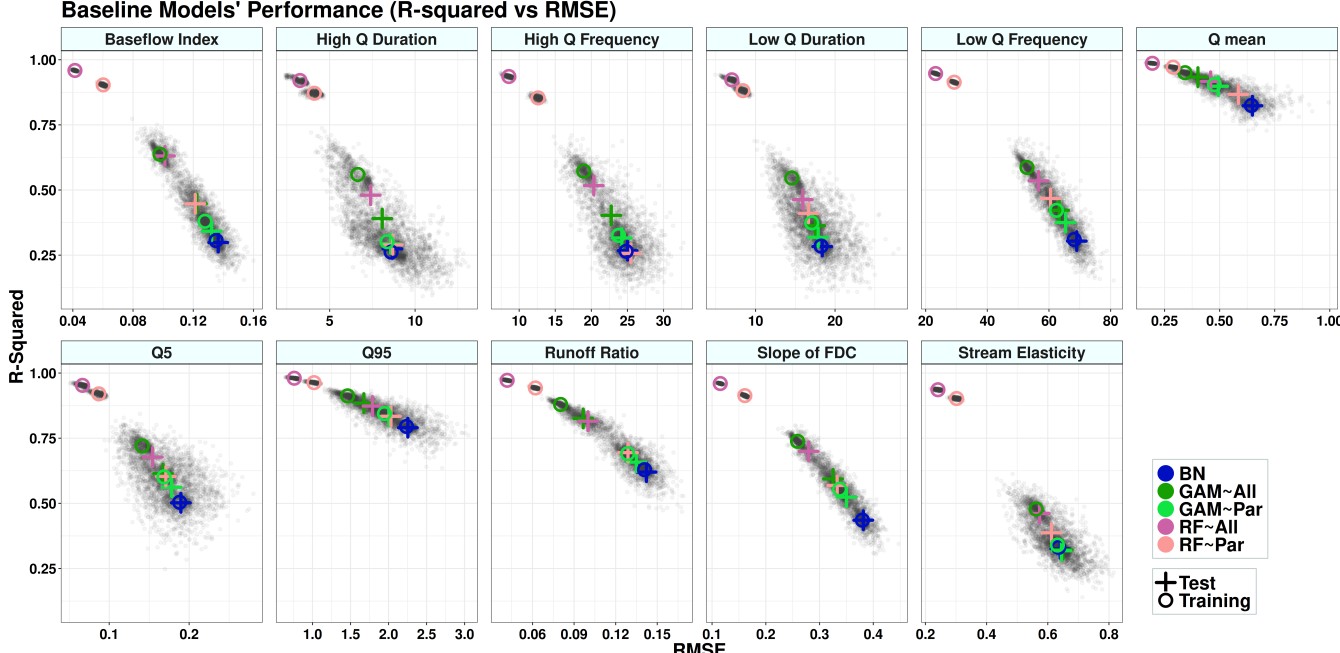

**Figure 4.** Performance of the models: R-squared vs. RMSE. Each coloured circle and cross represent the centroid of a set of 500 data points (grey dots) generated from the models' execution. Circles indicate the training results, and crosses indicate the test results. In the legend, "All" refers to using all variables as predictors (non-causal model), and "Par" refers to using only parent variables as predictors (causal model). BN refers to the Bayesian Network, GAM refers to the Generalized Additive Model, and RF refers to Random Forest.

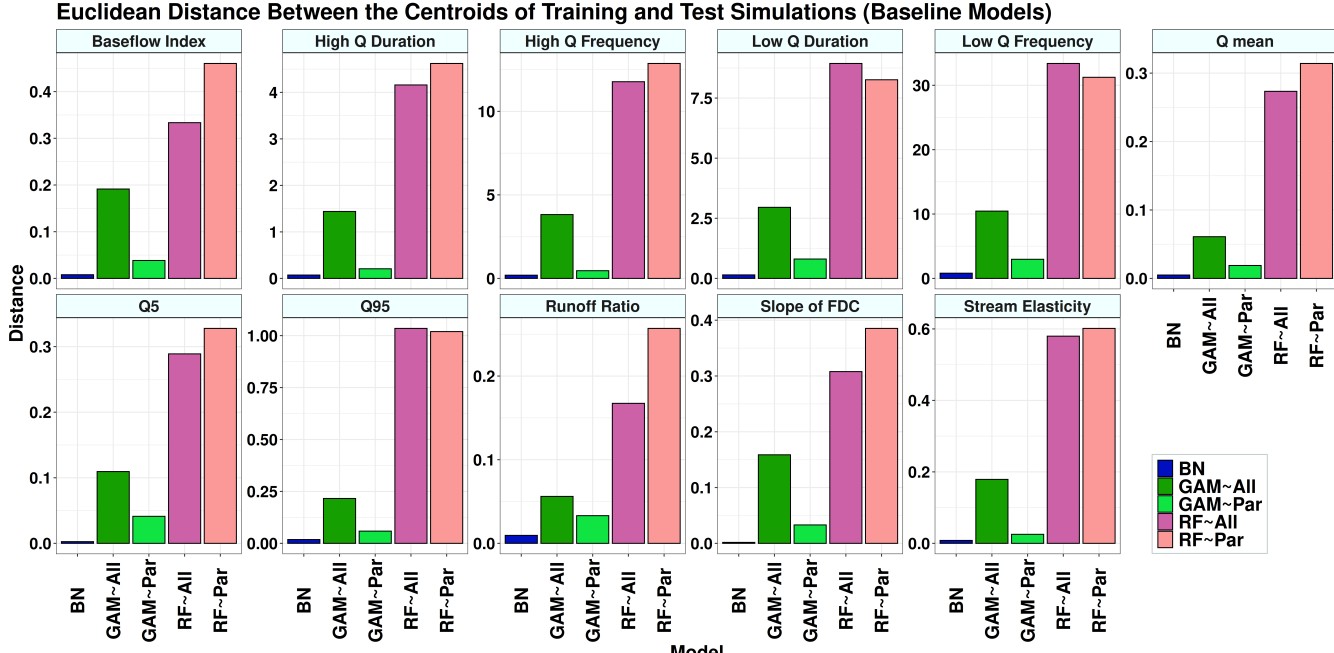

**Figure 5.** The Euclidean distance between the centroid points of training and test simulations in Fig. 4. In the legend, "All" refers to using all variables as predictors, and "Par" refers to using only parent variables as predictors. BN refers to the Bayesian Network, GAM refers to the Generalized Additive Model, and RF refers to Random Forest.

We see that when the training set is large, the accuracy of the non-causal models is higher (GAM~All and RF~All). However, this pattern might not be the same if the size of the training set is reduced. Testing the models in different environments with different properties and sizes can help us understand how these models perform. In this study, environments are clusters of catchments, defined according to each category of attributes (Table 3) that result in homogeneous hydrological properties. The selected variables for the DAG structure and analysis are assumed to be the same, both with and without clusters. However, in the analysis based on clusters, the model's parameterization and predictions are derived from a smaller subset of data compared to the baseline models. The direct causes of signatures are assumed to be the same across all clusters. Therefore, causal models are assumed to result in robust prediction in different environments. This idea is investigated in the following sections.

## 4.4 The performance of models across different clusters (Sub-models)

The results of this simulation indicate different models' behaviours across clusters, which are shown in Fig.7, Table 5, tables in Appendix A, and the sections 2 and 3 of the Supplementary Materials. According to the results, GAM~All shows high accuracy during training across most clusters, but performs poorly during testing. The distance between training and test for GAM~Par is lower than GAM~All in all clusters, and in most cases. This may be due to overfitting in GAM~All when the sample size is small, resulting in its performance being statistically insignificant from, or in most cases lower than, GAM~Par

across all environments in the test mode. On the other hand, RF∼All shows the highest performance in most cases in both training and testing modes. However, in many cases, RF∼Par performs comparably to RF∼All, despite using significantly fewer predictors. In the case of the BN model, which is linear, it generally exhibits the lowest accuracy compared to GAM and RF. However, it also shows the smallest drop in accuracy between training and testing simulations (see Supplementary Materials Section 2, Table S1, and Table S2). The simulation results for each runoff signature are discussed in the following sections.

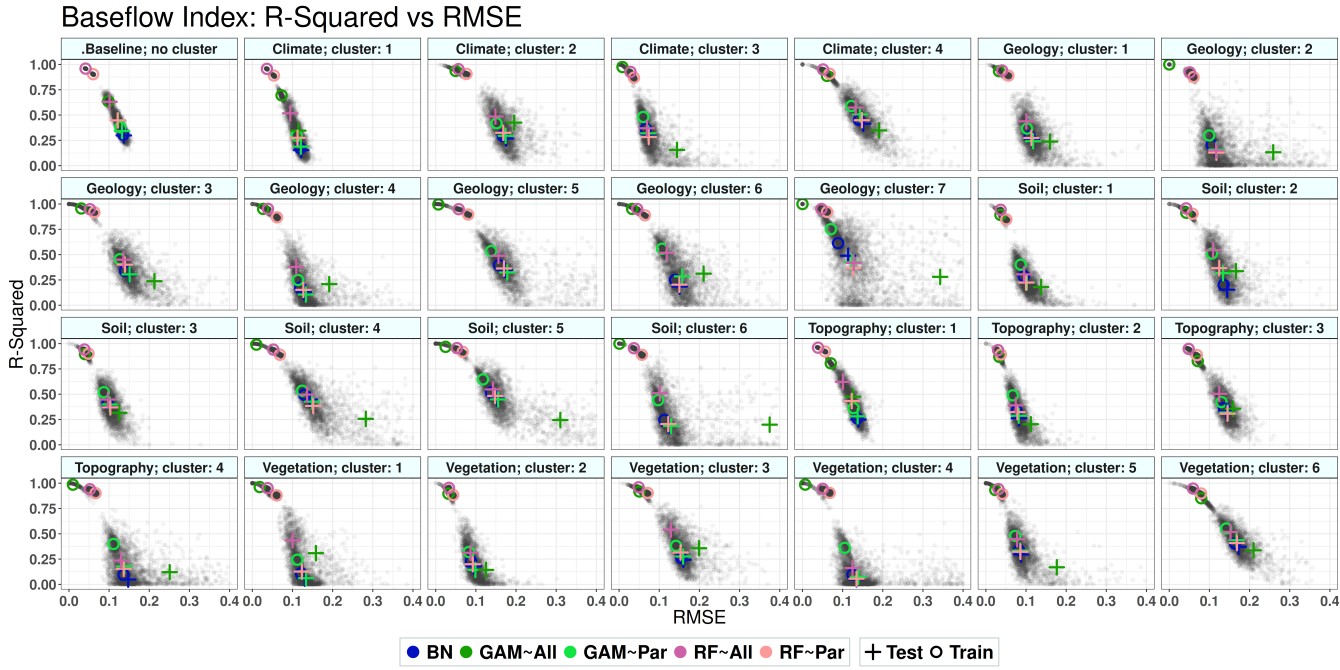

**Figure 6.** Performance of the models for baseflow index: R-squared vs. RMSE. Each coloured circle and cross represents the centroid of 500 data points (grey dots) generated from the models' execution. Circles indicate the training results, and crosses indicate the test results. In the legend, "All" refers to using all variables as predictors (non-causal model), and "Par" refers to using only parent variables as predictors (causal model). BN refers to the Bayesian Network, GAM refers to the Generalized Additive Model, and RF refers to the Random Forest. The results for other signatures are provided in supplementary materials.

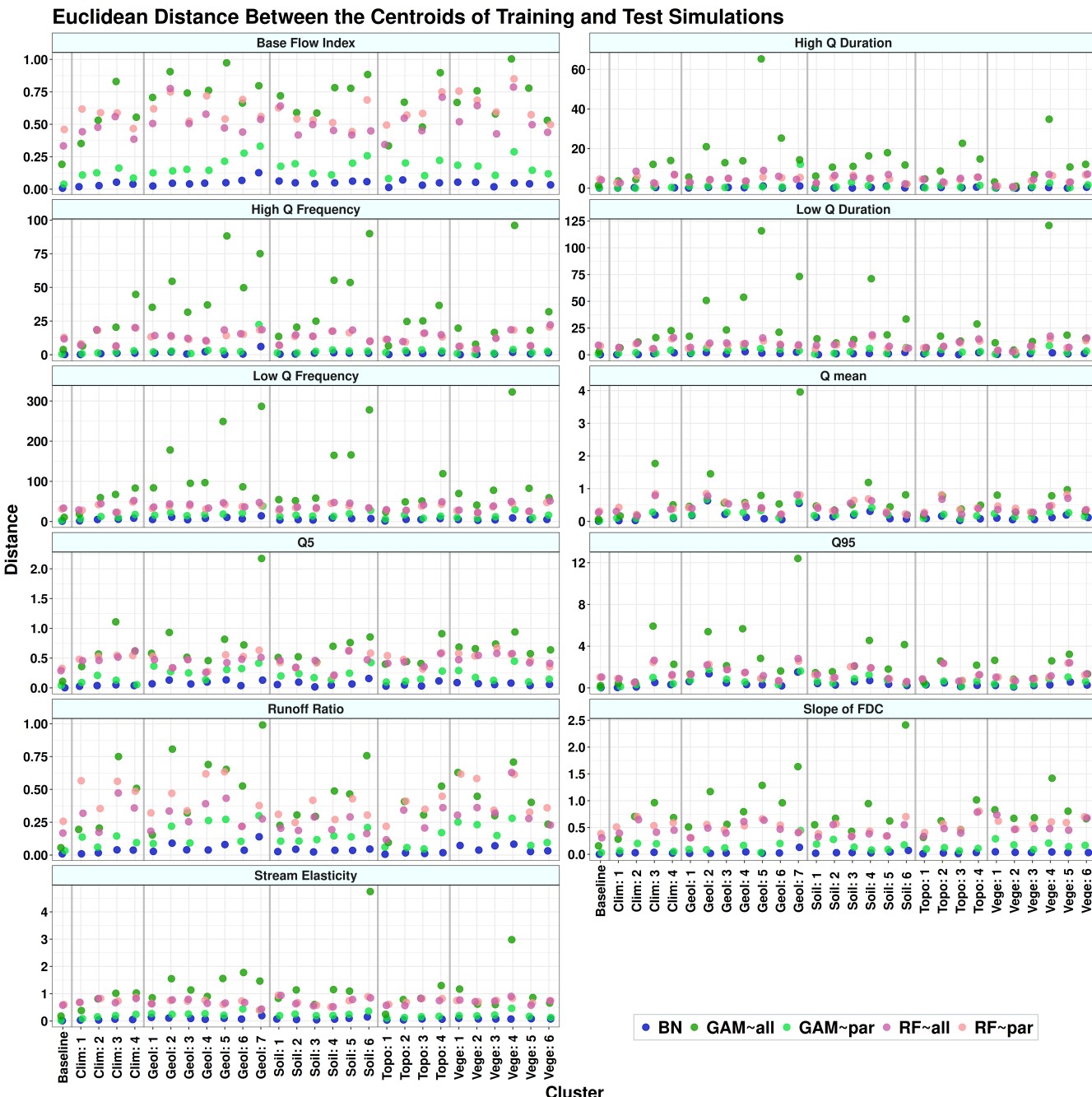

**Figure 7.** The Euclidean distance between the training and test simulation for runoff signatures across different environments for each sub-model. In the legend, "All" refers to using all variables as predictors (non-causal model), and "Par" refers to using only parent variables as predictors (causal model). BN refers to the Bayesian Network, GAM refers to the Generalized Additive Model, and RF refers to the Random Forest. On the x-axis, Baseline means simulation without any clustering and is done for all 671 catchments. Clim stands for climate, Geol for geology, Topo for topography and Vege for vegetation. The numbers in front of these names on the x-axis represent the clusters' numbers.

1. **Baseflow Index:** The four parents of this signature belong to climate, vegetation, and geology categories (Table 4). The identified causal parents exhibit both high statistical significance and strong edge strength (Table 4 and Fig. S23). The models in the climate, topography and some clusters of soil groups perform well compared to the baseline (Fig. 6). Although RF~All demonstrates the best performance, in most cases, the difference between the accuracy of the RF~All and RF~Par in the test set is negligible, for example, in soil category cluster numbers 3, 4, and 5, or Geology 2 and 7 (Fig. 6; Table S2). BN has the lowest distance between training and test (Fig. 7). The decrease in R-squared made by GAM~Par is improved from a -24% drop for the baseline model to +9% for geology, +27% for soil, and +4% for vegetation categories in the sub-models (Table 5). For the climate and topography categories, the accuracy drop caused by using GAM~Par is 6% and 4% smaller in the sub-models compared to the baseline model. The use of causal RF (RF~Pa) results in a 25% drop in accuracy in the baseline model. This reduction becomes more pronounced in the sub-models, except for the soil category, where the accuracy drop is 2% smaller than in the baseline.

2. **High Flow Duration:** This signature has two causal parents belonging to climate and vegetation categories (Table 4). The two causal parents of the high flow duration exhibit both high statistical significance and strong edge strength (Table 4 and Fig. S25). The models perform well across some clusters of the soil and geology categories compared to the baseline (Fig. S26). GAM~All shows very high accuracy in the training sets, in some cases higher than random forest, and a significant drop in accuracy in the test sets (Fig. S26). In addition, the distance between training and the test is higher than GAM~Par in all cases (Fig. 7). The causal GAM models show robust performance for all environments (Table S1). The distance between training and test simulation in RF~Par is mainly smaller than RF~All (Fig. 7). In many cases, the difference between causal and non-causal RF models is negligible (Table S2, Fig. S26). Although BN shows less accuracy compared to GAM and RF, it outperforms these models in some soil and geology clusters. The use of causal parents as predictors leads to a 26% and 40% drop in R squared in the GAM and RF simulations, respectively. Although the causal models use only two predictors compared to 22 in the non-causal models, the inclusion of causal parents increases the accuracy of GAM by up to 50% in the geology, soil, topography, and vegetation categories. Additionally, they help reduce the accuracy drop of RF in the sub-models relative to the baseline model.

3. **High Flow Frequency:** This signature has two parents belonging to the climate and geology categories (Table 4). Unlike high flow duration, the causal parents of the high flow frequency does not show high statistical significance with p-value of 0.24 and 0.06 for low precipitation frequency and geological porosity, respectively; however, they show acceptable edge strength with 79% for the former and 52% for latter (Table 4 and Fig. S27). Models perform well across some clusters of climate, soil, and geology categories. However, there is no single category within which all models outperform the others (Fig. S28). For instance, the models perform well in Vegetation Cluster 5 (Fig. S28), which are catchments with a high percentage of vegetation cover (Fig. 2). In general, GAM~All does not show acceptable performance in the test set, and its accuracy in many cases is lower than linear BN (Fig. S28). However, GAM~Par demonstrates a better performance by reducing the distance between training and test simulations (Fig. 7) and increasing accuracy compared to GAM~All across all clusters (Fig. S28; Table S1). Similarly, RF~Par decreases the distance between the training and

test across most of the clusters, although for the baseline models, this distance is smaller for RF∼All than RF∼Par (Fig. 7). However, the difference between RF∼All and RF∼Par is negligible in only three environments, namely, Geology 5, 7 and soil 5. For the rest of the environments, RF∼All is more accurate (Table S2). In most cases, the accuracy of GAM∼Par is higher than RF∼Par. Using causal parents leads to a decrease in the R-squared in the baseline model for both GAM and RF by -20% and -50%, respectively. While this behaviour remains for RF across sub-models, the causal parents increase the accuracy of the GAM by up to 65% (Table 5).

4. **Low Flow Duration:** This signature has three parents belonging to climate and soil categories (Table 4). All causal parents of the low flow duration exhibit both high statistical significance and strong edge strength (Table 4 and Fig. S29). Training and test simulation performed well across all topographic clusters except for cluster number 4, where catchments have high elevations (Fig. 2 and Fig. S30). The signature also shows high predictability in clusters with high precipitation intensity (Climate Cluster 3) and clusters with low soil porosity (Soil Cluster 2) or clusters with low maximum water content (Soil Cluster 3). GAM∼Par performs better in different clusters than GAM∼All by reducing the distance between training and test simulation and increasing the model's accuracy (Table S1). This distance is almost the same across clusters for RF∼Par and RF∼All and, in some cases, smaller for RF∼Par and in most environments, the difference between RF∼Par and RF∼All is not significant (Table S2). Using causal parents results in a decrease in R-squared values of approximately 10% for both GAM and RF. However, causal parents improve the accuracy of GAM across all categories by up to 85%, whereas for RF, the pattern is reversed, with accuracies dropping by up to 15%.

5. **Low Flow Frequency:** This signature has five parents, two belonging to climate, one to vegetation, and two to geological categories (Table 4). Three out of the five causal parents of low flow frequency exhibit both high statistical significance and strong edge strength. However, despite the relatively high edge strengths, 71% for forest fraction and 66% for geological porosity, the p-values from the likelihood ratio test are large (Table 4 and Fig. S31). Models perform well across most clusters of climate and topography categories (Fig. S30). In most cases, GAM∼All performs poorly compared to GAM∼Par (Table S1). The difference between training and testing is significantly reduced in GAM∼Par. This distance is also reduced in RF∼Par and, in many cases, the performance of the BN, GAM∼Par, RF∼Par, and RF∼All are comparable across most clusters (Fig. S32). Causal parents lead to a drop in the R-squared of both GAM and RF models. This pattern is the same for RF across the sub-models; however, it leads to improving the accuracy of the GAM by up to 33% (Table 5).

6. **Mean Daily Runoff:** The five parents of the mean daily runoff belong to climate, topography, vegetation, and geology categories (Table 4 and Fig. S33). All five causal parents of mean daily flow exhibit both high statistical significance and strong edge strength, except for forest fraction, which has a p-value of 0.06 in the likelihood ratio test (Table 4 and Fig. S33). This signature is the most predictable runoff signature. All models perform well across all clusters; however, unlike other signatures, BN and GAM models outperform RF in most cases, for example, Geology Cluster 2 (Fig. S34). In most cases, the difference between training and test simulations is smaller when using parents, which shows the benefits of using causal parents. In addition, the difference in model accuracy between simulations using only causal parent (∼Par)

and those using all variables (∼All) is negligible across almost all clusters, especially, for the GAM (Table S1). For mean daily flow, the reduction in R-squared resulting from using causal parents as predictors is minimal, even in the baseline models, approximately -8% for GAM and -5% for RF. While the accuracy drop in RF increases to as much as -15%, using causal parents improves GAM accuracy by up to 12%.

7. **Low Flow (Q5):** The five parents of low flow belong to climate, geology, and topography categories (Table 4 and Fig. S35). Among the causal parents of low flow, which generally exhibit both high statistical significance and strong edge strength, mean slope and geological porosity have higher p-values in the likelihood ratio test compared to the other parents, although their edge strengths remain high (Table 4 and Fig. S35). The models' test results are comparable to the baseline models in Geology Cluster 2 and 4 and Soil Cluster 2 and 4 (Fig. S36). GAM∼All is outperformed by GAM∼Par and other models in test simulation (Fig. S36, Table S1). As shown in Fig. S36, models perform well across the topographic category. The difference between training and test simulation is improved in GAM∼Par compared to GAM∼All. This distance for RF∼Par is smaller than RF∼All across half of the clusters (Fig. 7) and the difference between causal and non-causal RF models is negligible for most environments (Table S2). BN has the smallest difference between training and testing, and its performance is comparable to GAM and RF in most cases. Using parents as predictors increases the accuracy of GAM in the climate, geology, soil, and vegetation categories by 13%, 34%, 41%, and 31% (Table 5). For the RF, the drop in accuracy remains consistent across baseline and sub-models by around -10% to -15% Table 5).

8. **High Flow (Q95):** High flows are among the most identifiable signatures. According to the obtained DAG, high flow is controlled by five causal parents belonging to vegetation (land cover), climate, and topography category (Table 4 and Fig. S37). Among all the causal parents of high flow, precipitation seasonality has the highest p-value (0.15), despite its strong edge strength (98%). On the other hand, the catchment area shows the lowest edge strength (35%) and a relatively high p-value (0.06). The rest of the causal parents of this runoff signature exhibit high statistical significance and high edge strength (Table 4 and Fig. S37). The models showed high accuracy across most clusters. Unlike other signatures, the the accuracy of RF∼All and RF∼Par models, which are the most accurate overall, are comparable to GAM and BN in certain cases (Fig. S37). The difference between training and test simulations is improved in all clusters when using parents for GAM, except for climate cluster 1 and 2 as well as topography cluster 1(Table S1). The drop in the models' accuracy, caused by using the causal parents, is smaller compared to the other signatures. The R-squared is improved among geology, soil, and vegetation categories for GAM models (Table 5).

9. **Runoff Ratio:** Runoff ratio has four parents belonging to climate, geology, and topography categories (Table 4 and Fig. S39). Although catchment area has a high p-value as a causal parent of stream elasticity, its edge strength is 100%. All other causal parents of the runoff ratio exhibit both high statistical significance and strong edge strength (Table 4 and Fig. S39). The models perform well across topographic and soil clusters, and models are more robust across those environments (Fig. S39). Causal models show negligible difference between training and test simulations for almost all

clusters for GAM but not for RF (Fig. 5, Table S2). The difference between R-squared values is significantly lower across categories than the baseline models, especially in geology and soil categories (Table 5).

10. **Slope of Flow Duration Curve:** All three parents of the slope of the flow duration curve belong to climate category (Table 4, Fig. S41). All causal parents of the slope of the flow duration curve exhibit both high statistical significance and relatively strong edge strength (Table 4 and Fig. S41). Models in topographic clusters performed well except for the cluster 4, where there are catchments with a high elevation and steep slopes. RF∼Par and GAM∼Par perform almost the same across most of the clusters. In most cases, GAM∼Par reduced the difference between training and test simulations compared to GAM∼All (Table S1). The difference between RF∼Par and RF∼All is statistically significant except for two clusters, namely Geology Cluster 4 and Vegetation Cluster 1 (Fig. S42, Table S2). Using causal parents as predictors for the slope of the flow duration curve, increase the accuracy of the sub-models GAMs by 42%, 25%, and 33% for geology, soil, and vegetation categories; however, for the causal RF models lead to decrease in R-sqaured compared to the non-causal RFs (Table 5).

11. **Stream Precipitation Elasticity:** The five parents of this signature belong to climate, soil, topography, and vegetation categories (Table 4 and Fig. S43). Among the causal parents, snow fraction and forest fraction exhibit both high statistical significance and relatively strong edge strength. In contrast, the remaining causal parents have either relatively low edge strength or high p-values in the likelihood ratio test (Table 4 and Fig. S43). In most cases, the performance of RF∼All and RF∼Par models in the test mode is comparable (Fig. S44), especially in the geology cluster numbers 2, 3, 5, and 7 (Table S2). The same as other signatures, GAM∼All performs well only in the training simulation. The difference between causal and non-causal models is not statistically significant for the GAMs (Table S1). The distance between training and test simulation in GAM∼Par is smaller than GAM∼All. This pattern can be seen in only one-third of the clusters for RF models (Fig. 7). According to the Table 5, the causal parents lead to increase in accuracy of the GAM model by 17%, 75%, 14%, and 15% for climate, geology, soil, and vegetation categories The performance of RF∼All, RF∼Par and GAM∼Par are close and comparable in the test simulation (Fig. S43, Table 5, and Table S2).

**Table 5.** Comparison of R-squared values between causal and non-causal models presented as percentages. Negative values indicate a decrease in R-squared when using causal models compared to non-causal models. The R-squared values for each category are calculated using the weighted mean, with weights based on the proportion of catchments in each cluster relative to the total number of catchments. Red indicates a decrease in R-squared, while blue indicates an increase.

The values of R-squared can be found in Table A2 and Table A3.

| | Percentage of change in R squared made by using causal parents | | | | | | | | | | | |
| | Baseline | | Climate | | Geology | | Soil | | Topography | | Vegetation | |
| Signature | GAM | RF | GAM | RF | GAM | RF | GAM | RF | GAM | RF | GAM | RF |
|---|---|---|---|---|---|---|---|---|---|---|---|---|
| baseflow_index | -24.44 | -28.57 | -17.65 | -38.00 | 8.7 | -35.71 | 26.92 | -26.90 | -20 | -32.00 | 4.00 | -38.1 |
| high_q_dur | -25.64 | -39.58 | -15.15 | -30.23 | 50.00 | -19.57 | 26.92 | -23.81 | 15.38 | -32.50 | 17.39 | -28.95 |
| high_q_freq | -20.00 | -50.00 | -25.93 | -62.79 | 65.00 | -30.77 | 21.74 | -41.03 | 5.88 | -31.11 | 15.88 | -50.00 |
| low_q_dur | -11.11 | -10.87 | 11.54 | -17.65 | 85.00 | -7.89 | 7.41 | -14.71 | 0.00 | -14.29 | 18.18 | -23.53 |
| low_q_freq | -9.52 | -12.96 | 0.00 | -27.27 | 42.86 | -10.86 | 33.33 | -14.63 | -9.78 | -12.00 | 25.00 | -26.32 |
| q_mean | -3.23 | -5.43 | -11.84 | -15.79 | 12.33 | -8.75 | 7.5 | -4.71 | 4.88 | -10.71 | 1.37 | -12.00 |
| Q5 | -8.20 | -11.76 | 12.90 | -11.11 | 34.37 | -11.54 | 41.18 | -3.85 | -5.26 | -15.09 | 30.77 | -13.64 |
| Q95 | -5.68 | -4.60 | -9.23 | -12.50 | 28.33 | -1.37 | 14.29 | -2.47 | -9.23 | -12.50 | 4.76 | -10.45 |
| runoff_ratio | -20.48 | -15.85 | -38.33 | -35.38 | 13.46 | -23.19 | 1.61 | -17.33 | 0.00 | -17.11 | -9.09 | -23.81 |
| slope_FDC | -11.86 | -18.57 | -12.50 | -33.96 | 42.42 | -12.96 | 25.00 | -15.00 | -4.44 | -23.21 | 33.33 | -22.73 |
| stream_elast | 0.00 | -15.22 | 16.67 | -24.14 | 75.00 | -16.67 | 14.29 | -15.62 | -3.70 | -27.03 | 15.00 | -23.33 |

Fig. 8 displays the rankings of the overall performance of models across different environments for all signatures. RF∼All achieved the highest overall accuracy in the baseline mode, where the whole dataset is used. In contrast, BN in the baseline mode ranks 11th, which is the lowest among all models in this mode. This suggests that BN is either not sensitive or only weakly sensitive to sample size. For the other models, the baseline mode consistently ranks among the top 10 in terms of performance. Examining the top 10 rankings of the models across all environments reveals distinct patterns. For BN, the top-performing clusters include 2 from the climate category, 3 from geology, 2 from soil, and 1 from topography. For GAM∼Par, the top 10 includes the baseline, 1 cluster each from climate and topography, 2 from geology and vegetation, and 3 from the soil category. For GAM∼All, the top ranks comprise the baseline, 3 clusters from climate, and 2 clusters each from soil, topography, and vegetation, with none from geology. In the case of RF∼Par, the top 10 includes 1 cluster each from the baseline, climate, topography, and vegetation, 2 from geology, and 4 from the soil category. Lastly, for RF∼All, the top-performing clusters include 1 from the baseline, 1 cluster each from geology and topography, 2 from climate and vegetation, and 3 from the soil category. Overall, across the top 10 rankings of all models, clusters associated with the soil category appear most frequently, with 15 occurrences. This is followed by the climate category with 9 occurrences, geology and vegetation with 8 each, and topography with 6. Among all environments, Climate 4, followed by Vegetation 6, are the catchment groups where all models consistently achieve high rankings. Catchments in both climate 4 and Vegetation 6 are characterized by relatively

low precipitation, high evapotranspiration, and low maximum leaf area index. On the other hand, Vegetation 4, followed by Topography 4, are the environments where catchments consistently exhibit the lowest model performance. Catchments in Vegetation 4, which are mostly located in the southern and southeastern U.S., show high variability in forest fraction and maximum leaf area index. In contrast, catchments in Topography 4 are primarily situated in the Rocky Mountains and are characterized by high elevation and steep slopes.

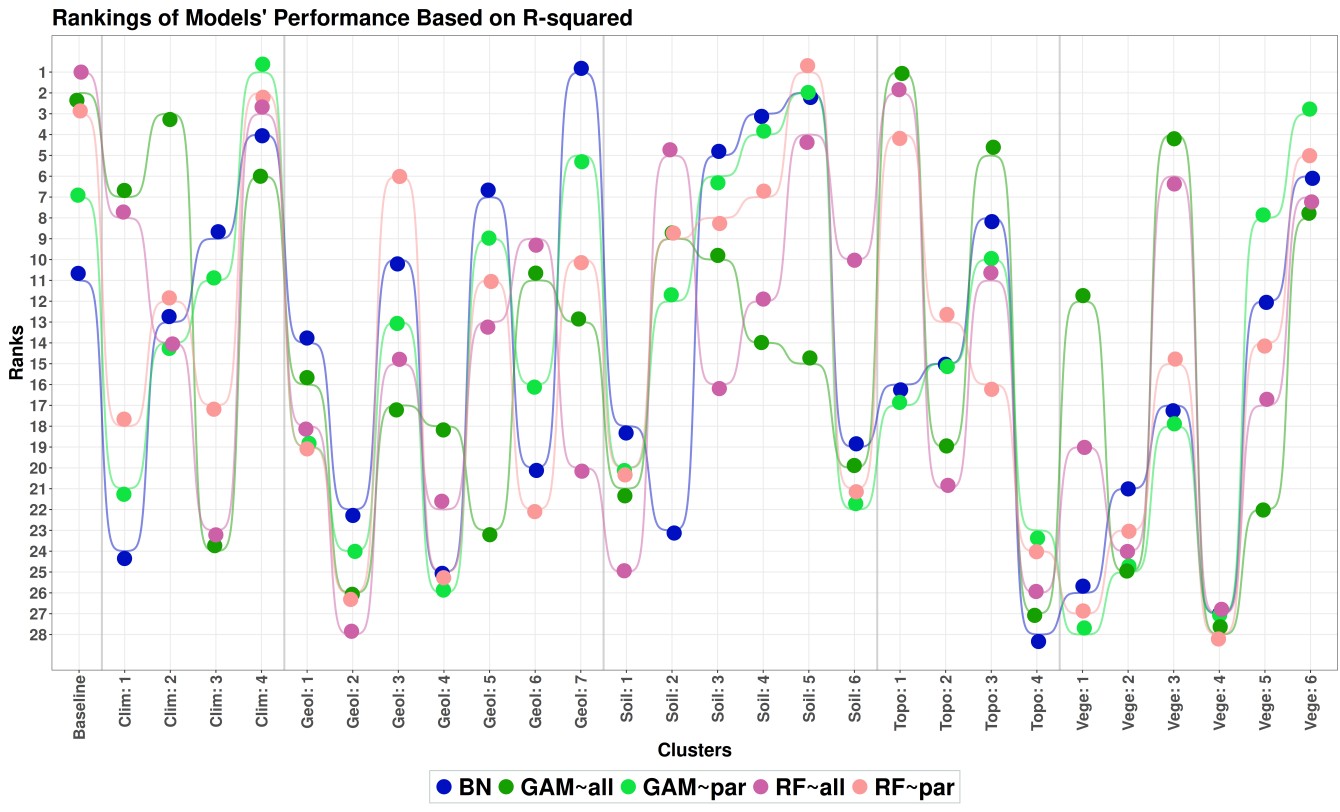

**Figure 8.** Rankings of model performance based on R-squared values obtained from evaluating their accuracy in predicting all signatures within each cluster. On the x-axis, Clim stands for climate, Geol for geology, Topo for topography and Vege for vegetation.

## 5    Discussion

The aims of this study have tried to (1) recover the causal graph, represented as a Directed Acyclic Graph (DAG), from catchment attributes, climate characteristics, and runoff signatures; (2) predict runoff signatures using their causal parents as well as all variables in the DAGs; and (3) compare the predictive performance of models using only causal parents (which is an independent causal mechanism) versus those using all available variables.

PC-stable algorithm (Colombo and Maathuis, 2014) is used to uncover the underlying causal relationships between runoff signatures, catchment attributes, and climate characteristics. To improve the plausibility of the resulting graph, background

knowledge is applied before running PC. Applying background knowledge reduces the risk of spurious or false positive edges, decreases the number of equivalence classes, and improves the stability of the learned causal graph (Meek, 1995; Spirtes et al., 2001; Perković et al., 2017; Bang and Didelez, 2025). The background knowledge is applied by blocking the implausible edges. In hydrological systems, many causal directions are well understood from process-based reasoning and can be used to restrict the search space of the algorithm. By enforcing these structural constraints (e.g., forbidding reverse causality from runoff to climate), the stability of the learned causal graph is improved, and it allows us to derive a complex causal graph for the catchment area.

To mitigate the rate of false positives and negatives (Type I and II errors) in the PC algorithm due to our limited sample size (Li and Wang, 2009), we start the causal discovery analyses by applying a relatively lenient significance threshold (Kalisch and Bühlman, 2007). This threshold increases the risk of false-positive edges. Therefore, to evaluate the quality of the fit, we assess the significance levels of dependencies between adjacent nodes and their corresponding strengths (Petersen et al., 2021). We perform a heuristic evaluation using cubic spline regression and the likelihood ratio test, which allows for examining nonlinear dependencies between adjacent nodes, particularly given that real-world hydrological systems often exhibit nonlinear relationships (Kirchner, 2024). To further assess the strength and stability of the inferred links, we perform bootstrap resampling and learn the network structure 1000 times by the PC algorithm using a more stringent significance level (Scutari and Nagarajan, 2013). This approach helps us investigate potential false-positive edges (Type I errors) in the DAG obtained using the lenient significance threshold. Among all the inferred edges in the resulting DAGs, the link between low precipitation frequency and maximum leaf area index exhibits significantly low strength.

The causal parents identified by the PC algorithm align with the underlying physical processes for most of the signatures. For example, according to PC results, snow fraction drives the baseflow index, consistent with runoff-generating mechanisms during spring and summer (Gentile et al., 2023). In addition, vegetation and geological variables, which contribute to infiltration and groundwater flow, are causal parents of the baseflow index (Gnann et al., 2019). For high flows (Q95), drivers include precipitation characteristics (mean, seasonality, and frequecy), vegetation cover, and catchment area. This suggests that precipitation intensity, often driven by seasonality, influences runoff-generating mechanisms like the infiltration excess process (Nanda et al., 2019). Area and vegetation cover also affect the time concentration and the magnitude of high flows in the catchment area (Sultan et al., 2022). In regions with high mean precipitation and low seasonality, saturation excess runoff mechanisms dominate high flows. Additionally, the PC results for low flows (Q5) includes two variables belong to the geology, two climate, and one topography. Low flows are strongly governed by geological variables in addition to climate and topography (Laaha and Bloeschl, 2006; Giuntoli et al., 2013). However, the causal parents identified by the PC method for the slope of the flow duration curve include only climate variables. While this flow signature is strongly influenced by catchment storage, shaped by geology, topography, and land cover (Dey et al., 2024), the method failed to capture these non-climatic influences. The identified causal parents of high flow frequency include high precipitation frequency and geological porosity. However, land cover is also recognized as a key driver of this flow signature (Zabaleta et al., 2018), though it was not captured by the causal discovery method. Therefore, while the PC algorithm identifies physically meaningful causal parents for most signatures, it occasionally fails to capture expected parent variables from certain categories that are known to influence those

signatures. Furthermore, the derived causal parents by the PC are not necessarily the highest correlated variables with the runoff signatures. This highlights that there can be strong causal relationships between variables even when their statistical associations Gao et al. (2023).

The obtained DAGs indicate that topographic variables drive climate, vegetation, geological and soil variables of hydrological systems at the catchment scale. Also, they show that climate attributes influence all runoff signatures, a finding supported by various studies (E.g. Jehn et al. (2020); McMillan et al. (2022)). Models perform well across soil and climate clusters for most signatures with consistent high accuracy rankings (Fig. 8). However, in Vegetation Cluster 4 and Topography Cluster 4all models struggle to predict signatures accurately. In the catchments of Vegetation Cluster 4, forest fraction varies almost uniformly between 0 and 1, and as a result, the maximum leaf area index exhibits a similar pattern (Fig. 2). This high variability, combined with the small sample size of 69 catchments, leads to poor model performance (Table 3). On the other hand, catchments in the Topography Cluster 4 are characterized by high elevation and low precipitation. The low prediction accuracy in this cluster aligns with Viglione et al. (2013), who observed a decline in prediction model performance in arid catchments. Signatures prove to be more predictable in clusters characterized by high precipitation and low elevation, such as those in Climates 1 and 3. This indicates that even in catchments with low precipitation, the transfer of information from precipitation to runoff remains the predominant driver compared to other mechanisms (Neri et al., 2022). According to Fig. 8, models achieve high accuracy scores in regions with high precipitation, such as Topography 1 and Soil 5. The prediction results indicate that independent variables derived from causal discovery, such as topographic variables, can serve as effective criteria for catchment classification.

The causal GAM model, GAM~Par, outperforms the non-causal GAM~All in most environments during testing, despite the latter exhibiting higher accuracy in the training mode. GAM~All achieves its best performance in the baseline mode, where the entire dataset is used for simulation. However, even in these baseline cases, its performance remains comparable to that of GAM~Par. In the case of RF models, RF~All achieves the highest accuracy across most environments compared to the other models; however, the performance of the RF~Par is still comparable to RF~All.. The difference in performance between RF~All and RF~Par is statistically significant in most cases, as determined by a non-parametric permutation test (see the Supplementary Materials Section 3). However, this significance does not merely represent the magnitude of difference between the models. The results of the test also depend on the spread of results across the R-squared and RMSE space. For example, two models might differ in average accuracy, and if their performance varies widely during bootstrapped training and testing, the difference may not be statistically significant. Conversely, even a small difference in accuracy can be statistically significant if the performances of the models are consistently stable. Despite BN having lower accuracy than GAM and RF, it shows the smallest difference between training and test results across all cases. This consistency may be due to the BN structure, which relies on conditional dependencies derived from the causal relationships between variables, although further investigation is needed. The difference between causal and non-causal RF models is mostly statistically significant across clusters of the baseflow index, high flow frequency, low flow frequency, mean daily flow, runoff ratio, the slope of the flow duration curve, and the stream precipitation elasticity. For signatures where the difference is insignificant, using causal parents can enhance model parsimony by reducing the number of predictors, improve robustness by maintaining accuracy across

environments comparable to non-causal models, and minimize accuracy reduction between the training and testing phases. This pattern holds for GAM∼Par across most clusters, and for RF∼Par across the majority of clusters related to low flow duration, high flow duration, low flows, and high flows.

Finally, our results show that causal discovery enhances the representation of physical systems, making models more interpretable and parsimonious, as emphasized by Runge et al. (2019a) and Reichstein et al. (2019). However, there is still room for further investigation, as the causal graphs obtained using the proposed methods require deeper analysis. In this study, we used the PC algorithm to identify the causal parents of runoff signatures. Other methods are specifically designed to target causal parents of a given variable, such as Invariant Causal Prediction (ICP), and could offer additional insights (see Peters et al. (2016); Heinze-Deml et al. (2018b); Kook et al. (2024)). The insights gained from causal discovery can not only improve the understanding of hydrological systems at the catchment scale, but also lead to more informed modelling practice (Slater et al., 2024). However, we still need theoretical developments to quantify the stability and robustness of the uncertainty of such a model, particularly when combined with machine learning and classification algorithms (Herman et al., 2015; Singh et al., 2015; AghaKouchak et al., 2022).

## 6 Conclusions

This study investigates the application of causal discovery to represent the causal interconnections between variables in hydrological systems. The PC algorithm is used to identify the causal links between catchment attributes, climate indices, and 11 runoff signatures, producing a Directed Acyclic Graph (DAG) for each signature. DAGs reveal the connections between variables, including the direct causes (parents) of the target signatures. Three prediction models, BN, GAM, and RF, in five different settings, namely BN, GAM∼Par, GAM∼All, RF∼Par and RF∼All, are used to predict runoff signatures. These models are executed on the entire dataset as well as 27 clusters, with each configuration undergoing 500 random samplings of training and test sets, resulting in a total of 140,000 model executions. BN directly utilizes the DAG structure for prediction, while GAM and RF predict the target variable both by using all the variables in the DAG and by using only the causal parents (the variables that, together with the target variables, form the independent causal mechanism). Each model is run 500 times with random sampling of training and tests for each run. The dataset is then grouped into different clusters based on attribute categories. The clusters serve as new environments to train and test the models, allowing for an assessment of model performance when using causal parents as the explanatory variables. The major outcomes of this research are as follows:

– The causal parents of the signatures identified by the PC algorithm do not always align with the most influential variables determined by correlation and variable importance analysis. This suggests that strong correlations may result from confounding variables, and causal relationships do not always coincide with high variable importance. This point can impact the robustness of prediction models, especially when the same set of predictor variables is used across diverse environments with varying characteristics.

- BN shows the smallest decrease in accuracy between the training and test samples, demonstrating high transferability. The accuracy of the models is not sensitive to the training sample size and shift in the distribution of predictors. This indicates that $P(\text{Effect} \mid \text{Cause})$ remains consistent across environments. Although BN's overall accuracy is lower than that of the nonlinear GAM and RF models, it outperforms RF in predicting mean daily runoff and high flows across different environments (clusters).

- Using causal parents helps mitigate the overfitting problem and improve the robustness in prediction models, particularly in GAM, when the size of the training set is small.

- The high accuracy of non-causal models, GAM~All , in the baseline scenarios may be attributed to over-fitting or spurious relationships. This is supported by their reduced accuracy in environments with smaller training sets, highlighting a lack of robustness compared to causal models, which maintain higher reliability under such conditions.

- Signatures are most predictable when causal and non-causal models are trained on catchments with homogeneous soil properties.

- Independent variables identified through causal discovery can determine groups of catchments where prediction models exhibit consistent performance. For instance, topographic variables are among the independent variables in this context since all models perform consistently well in clusters 1, 2, and 3, and less effectively in cluster 4. This is also the case for the soil and the climate categories where their variables are mostly independent of the other categories. This information helps identify environments where training models achieve higher accuracy, reduced uncertainty, and greater robustness.

- Causal inference methods contribute to improving prediction models' parsimony, interoperability and robustness in hydrological systems.

In conclusion, causal models maintain acceptable accuracy across environments with varying distributions of explanatory variables (covariates). The DAGs obtained from causal discovery enhance the interpretability of prediction models and offer more informed clustering criteria, which is valuable for regionalization purposes. This study focuses on investigating the direct causes of runoff signatures and their effects on prediction accuracy, but other criteria for selecting predictors from the DAG variables could be explored. For example, investigating the effect of variables with different topological ordering on the target variable, such as root nodes, ancestors of the target variables, etc. In addition, different causal discovery methods may yield alternative DAG structures, which merit further investigation. This work offers an insight into the application of causal inference methods in understanding runoff-generating mechanisms in hydrological systems.

While causal inference analysis has been extensively explored in fields such as computer science and medicine, its applications in hydrology are still in their infancy. There is a broad range of potential uses for causal models in hydrology, from identifying the drivers of hydrological anomalies (Tárraga et al., 2024) to linking extreme events with their cascading societal impacts (AghaKouchak et al., 2023). As research in this area progresses, the application of causal inference methods is likely to lead to more accurate and robust predictive models, offering valuable insights into complex hydrological variability.

*Code and data availability.* The codes are available on the GitHub repository at https://github.com/abbasizadeh/Catchment-Causal-Discovery. The CAMELS attributes are available at https://gdex.ucar.edu/dataset/camels.html.

## Appendix A: The values of R squared and RMSE for the baseline models and R squared values for sub-models

## A1 R-squared and RMSE values for test simulations of baseline models in Fig. 4

**Table A1.** R-squared and RMSE values for test simulations of baseline models. The values are an average of 500 executions of each model.

| Signature | R squared (Test Set) | | | | | RMSE (Test Set) | | | | |
|---|---|---|---|---|---|---|---|---|---|---|
| | BN | GAM∼All | GAM∼Par | RF∼All | RF∼Par | BN | GAM∼All | GAM∼Par | RF∼All | RF∼Par |
| baseflow_index | 0.30 | 0.45 | 0.34 | 0.63 | 0.45 | 0.14 | 0.12 | 0.13 | 0.10 | 0.12 |
| high_q_dur | 0.27 | 0.39 | 0.29 | 0.48 | 0.29 | 8.67 | 8.08 | 8.57 | 7.40 | 8.69 |
| high_q_freq | 0.27 | 0.40 | 0.32 | 0.52 | 0.26 | 25.01 | 22.76 | 24.22 | 20.33 | 25.48 |
| low_q_dur | 0.28 | 0.36 | 0.32 | 0.46 | 0.41 | 18.39 | 17.50 | 17.88 | 15.94 | 16.66 |
| low_q_freq | 0.30 | 0.38 | 0.42 | 0.54 | 0.47 | 69.02 | 63.31 | 65.40 | 56.62 | 60.50 |
| q_mean | 0.82 | 0.93 | 0.90 | 0.92 | 0.87 | 0.65 | 0.40 | 0.49 | 0.46 | 0.59 |
| Q5 | 0.50 | 0.61 | 0.56 | 0.68 | 0.60 | 0.19 | 0.17 | 0.18 | 0.15 | 0.17 |
| Q95 | 0.79 | 0.88 | 0.83 | 0.87 | 0.83 | 2.25 | 1.68 | 2.00 | 1.79 | 2.03 |
| runoff_ratio | 0.62 | 0.83 | 0.66 | 0.82 | 0.69 | 0.14 | 0.10 | 0.13 | 0.10 | 0.13 |
| slope_FDC | 0.44 | 0.59 | 0.52 | 0.70 | 0.57 | 0.38 | 0.33 | 0.35 | 0.28 | 0.33 |
| stream_elast | 0.33 | 0.32 | 0.32 | 0.46 | 0.39 | 0.64 | 0.65 | 0.64 | 0.57 | 0.61 |

 **A2    R squared values used to calculate values in Table 5**

**Table A2.** The R-squared values of causal models for each category, which are calculated using the weighted mean. The weights are the ratio of the catchments in each cluster to the total number of catchments.

| | R squared values for causal models | | | | | | | | | | | | | | |
| | Climate | | | Geology | | | Soil | | | Topography | | | Vegetation | | |
| Signature | BN | GAM | RF | BN | GAM | RF | BN | GAM | RF | BN | GAM | RF | BN | GAM | RF |
|---|---|---|---|---|---|---|---|---|---|---|---|---|---|---|---|
| baseflow_index | 0.24 | 0.28 | 0.31 | 0.25 | 0.25 | 0.27 | 0.30 | 0.33 | 0.34 | 0.25 | 0.28 | 0.34 | 0.22 | 0.24 | 0.26 |
| high_q_dur | 0.24 | 0.28 | 0.30 | 0.35 | 0.39 | 0.37 | 0.33 | 0.33 | 0.32 | 0.26 | 0.30 | 0.27 | 0.23 | 0.27 | 0.27 |
| high_q_freq | 0.17 | 0.20 | 0.16 | 0.30 | 0.33 | 0.27 | 0.26 | 0.28 | 0.23 | 0.30 | 0.36 | 0.31 | 0.19 | 0.22 | 0.18 |
| low_q_dur | 0.24 | 0.29 | 0.28 | 0.40 | 0.37 | 0.35 | 0.30 | 0.29 | 0.29 | 0.43 | 0.41 | 0.42 | 0.23 | 0.26 | 0.26 |
| low_q_freq | 0.25 | 0.28 | 0.32 | 0.32 | 0.30 | 0.35 | 0.30 | 0.32 | 0.35 | 0.33 | 0.37 | 0.44 | 0.23 | 0.25 | 0.28 |
| q_mean | 0.61 | 0.67 | 0.64 | 0.80 | 0.82 | 0.73 | 0.83 | 0.86 | 0.81 | 0.83 | 0.86 | 0.75 | 0.72 | 0.74 | 0.66 |
| Q5 | 0.31 | 0.35 | 0.40 | 0.40 | 0.43 | 0.46 | 0.45 | 0.48 | 0.50 | 0.35 | 0.36 | 0.45 | 0.34 | 0.34 | 0.38 |
| Q95 | 0.58 | 0.59 | 0.56 | 0.75 | 0.77 | 0.72 | 0.77 | 0.80 | 0.79 | 0.58 | 0.59 | 0.56 | 0.65 | 0.66 | 0.60 |
| runoff_ratio | 0.30 | 0.37 | 0.42 | 0.58 | 0.59 | 0.53 | 0.62 | 0.63 | 0.62 | 0.66 | 0.68 | 0.63 | 0.46 | 0.50 | 0.48 |
| slope_FDC | 0.26 | 0.35 | 0.35 | 0.42 | 0.47 | 0.47 | 0.44 | 0.50 | 0.51 | 0.37 | 0.43 | 0.43 | 0.25 | 0.32 | 0.34 |
| stream_elast | 0.22 | 0.21 | 0.22 | 0.28 | 0.28 | 0.30 | 0.25 | 0.24 | 0.27 | 0.25 | 0.26 | 0.27 | 0.23 | 0.23 | 0.23 |

**Table A3.** The Rsquared values of non-causal models for each category which are calculated using the weighted mean. The weights are the ratio of the catchments in each cluster to the total number of catchments.

| | R squared values for non-causal models | | | | | | | | | |
| | Climate | | Geology | | Soil | | Topography | | Vegetation | |
| Signature | GAM | RF | GAM | RF | GAM | RF | GAM | RF | GAM | RF |
|---|---|---|---|---|---|---|---|---|---|---|
| baseflow_index | 0.34 | 0.50 | 0.23 | 0.42 | 0.26 | 0.46 | 0.35 | 0.50 | 0.25 | 0.42 |
| high_q_dur | 0.33 | 0.43 | 0.26 | 0.46 | 0.26 | 0.42 | 0.26 | 0.40 | 0.23 | 0.38 |
| high_q_freq | 0.27 | 0.43 | 0.20 | 0.39 | 0.23 | 0.39 | 0.34 | 0.45 | 0.19 | 0.36 |
| low_q_dur | 0.26 | 0.34 | 0.20 | 0.38 | 0.27 | 0.34 | 0.41 | 0.49 | 0.22 | 0.34 |
| low_q_freq | 0.28 | 0.44 | 0.21 | 0.39 | 0.24 | 0.41 | 0.41 | 0.50 | 0.20 | 0.38 |
| q_mean | 0.76 | 0.76 | 0.73 | 0.80 | 0.80 | 0.85 | 0.82 | 0.84 | 0.73 | 0.75 |
| Q5 | 0.31 | 0.45 | 0.32 | 0.52 | 0.34 | 0.52 | 0.38 | 0.53 | 0.26 | 0.44 |
| Q95 | 0.65 | 0.64 | 0.60 | 0.73 | 0.70 | 0.81 | 0.65 | 0.64 | 0.63 | 0.67 |
| runoff_ratio | 0.60 | 0.65 | 0.52 | 0.69 | 0.62 | 0.75 | 0.68 | 0.76 | 0.55 | 0.63 |
| slope_FDC | 0.40 | 0.53 | 0.33 | 0.54 | 0.40 | 0.60 | 0.45 | 0.56 | 0.24 | 0.44 |
| stream_elast | 0.18 | 0.29 | 0.16 | 0.36 | 0.21 | 0.32 | 0.27 | 0.37 | 0.20 | 0.30 |

*Author contributions.* **Hossein Abbasizadeh**: Conceptualization, Methodology, Coding and computation, Data visualization, Writing – review & editing, Funding acquisition. **Petr Maca**: Supervision, Conceptualization, Methodology, Writing – review & editing, Funding acquisition; **Martin Hanel**: Methodology, Writing – review & editing, Funding acquisition; **Mads Troldborg**: Conceptualization, Methodology, Writing – review & editing; **Amir AghaKouchak**: Conceptualization, Methodology, Writing – review & editing.

*Competing interests.* The authors declare that they have no conflict of interest.

*Acknowledgements.* This study was supported by the Internal Grant Agency of the Faculty of Environmental Sciences, Czech University of Life Sciences Prague (project No.2023B0026 and No.2024B0003) and the Ministry of Education, Youth and Sports of the Czech Republic (grant AdAgriF - Advanced methods of greenhouse gases emission reduction and sequestration in agriculture and forest landscape for climate change mitigation (CZ.02.01.01/00/22_008/0004635)). Computational resources were provided by the e-INFRA CZ project (ID:90254), 815    supported by the Ministry of Education, Youth and Sports of the Czech Republic.

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
