# Peer review of "Can causal discovery lead to a more robust prediction model for runoff signatures?"

_Hydrology and Earth System Sciences, 2024_

## Author Comment (AC1)

**Reply to Reviewer 1**

Thank you very much for taking the time to review our manuscript and for providing us with constructive feedback. We have carefully addressed all the points raised and will incorporate the necessary changes into the revised manuscript. Please find below the main points to be corrected:

1. **The title, the abstract, and the conclusions are not fully aligned. In the title authors mention to "prediction", in the abstract "interpretation", in the conclusion there are many points mixing the two topics.**

   We agree that the concepts in these sections are somewhat mixed and could be clarified. To address this issue, we have revised the abstract and conclusion to better align with the title, which emphasizes "prediction." We specifically highlighted how the study's results can contribute to improving prediction models.

   Here are the following changes we will make:

   **Lines 13-14:** *This study demonstrates the potential of causal inference techniques for predicting catchment responses by effectively representing the interconnected processes within hydrological systems in a more interpretable manner.*

   **From line 479:**

   *Three prediction models, BN, GAM, and RF, in five different settings, namely BN, GAM∼Par, GAM∼All, RF∼Par and RF∼All, are used to predict runoff signatures. These models are executed on the entire dataset as well as 22 clusters, with each configuration undergoing 100 random samplings of training and test sets, resulting in a total of 11,500 model executions.*

   **From line 485:**

   - *The causal parents of the signatures identified by the PC algorithm do not always align with the most influential variables determined by correlation and variable importance analysis. This suggests that strong correlations may result from confounding variables, and causal relationships do not always coincide with high variable importance. This point can impact the robustness of prediction models, especially when the same set of predictor variables is used across diverse environments with varying characteristics.*

   - *BN shows the smallest decrease in accuracy between the training and test samples, demonstrating high transferability. The accuracy of the models is not sensitive to the training sample size and shift in the distribution of predictors. This indicates that $P(\text{Effect} \mid \text{Cause})$ remains consistent across environments. Although BN's overall accuracy is lower than that of the nonlinear GAM and RF models, it outperforms RF in predicting mean daily runoff and high flows across different environments (clusters).*

   - *Using causal parents helps mitigate the overfitting problem and improve the robustness in prediction models, particularly in GAM, when the size of the training set is small.*

   - *The high accuracy of non-causal models, GAM∼All and RF∼All, in the baseline scenarios may be attributed to spurious relationships. This is supported by their reduced accuracy in environments with smaller training sets, highlighting a lack of robustness compared to causal models, which maintain higher reliability under such conditions.*

   - *In environments where the target signature is more difficult to predict, such as clusters of the geology category, using causal parents increases prediction accuracy.*

- *Independent variables identified through causal discovery can determine groups of catchments where prediction models exhibit consistent performance. For instance, topographic variables are among the independent variables in this context since all models perform consistently well in clusters 1, 2, and 3, and less effectively in cluster 4. This information helps identify environments where training models achieve higher accuracy, reduced uncertainty, and greater robustness. ~~The independent variables identified through causal discovery using DAGs can serve as reliable criteria for catchment classification. This is evident from the models performing consistently well in clusters 1, 2, and 3, while performing less effectively in cluster 4. This information improves model accuracy, reduces prediction uncertainty, and enhances consistency between training and test simulations.~~*

- *Causal inference methods contribute to improving prediction models'  parsimony, interoperability and robustness in hydrological systems .*

2. **Runoff signature is synonymous of "hydrological response" or "watershed response"? Maybe in the introduction this other common term could be mentioned just to better orient the reader.**

Runoff signatures represent distinct characteristics of a catchment's response. While "catchment response" and "runoff signature" are sometimes used interchangeably, this could create confusion for readers. In the revised sections, we will carefully use "catchment response" to ensure clarity and avoid any ambiguity.

3. **In the lines 103-113 it should be clarified which is the innovative contribution or the advancement compared to the previous literature accurately listed by the authors**

We appreciate this reviewer's remark. We have revised the section to clearly highlight the novelty of our work and provide more context. Specifically, we emphasized that the use of causal discovery to identify causal links between catchment attributes, climatic indices, and runoff signatures and integrating these findings into prediction models represents the key innovation of this research.

To our current knowledge, the proposed study is the first analysis that connects the causal models and catchment attributes. Therefore, we will change this section to:

*This study introduces a novel approach for predicting runoff signatures by integrating causal information into predictive models. To the best of our knowledge, causal inference techniques have not yet been applied for this purpose. Unlike previous studies that primarily rely on correlated-based features for predicting a specific catchment response, we take a step beyond mere correlation by focusing on causally relevant variables, specifically, causal parents. By integrating causal information into predictive models (GAM and RF), we aim to investigate whether it can enhance the prediction models' robustness, interpretability, and parsimony compared to models that do not utilize causal insights.  We assume that a specific characteristic of catchment response is directly influenced by a subset of correlated variables, known as causal parents, rather than by all correlated variables. ~~runoff signatures are causally influenced by a subset of variables, known as causal parents, rather than by all available variables. We adopt the Peter and Clark (PC) causal discovery method (Spirtes et al., 2001), which is a constrained-based causal discovery algorithm, to identify these causal relationships and to structure the BNs. Our objective is to investigate whether incorporating causal information can provide new insight into hydrological systems modelling, enhance the prediction models' robustness, and improve their parsimony.2) identify causal parents and network structure for each signature,~~ 3) execute models using both the causal parents (causal models) and all selected variables (non-causal models) for entire catchments and subset of catchments, 4) evaluate the robustness of the causal and non-causal models.*

4. **Section 3. Data are crucial for understanding the model application. In the Section 3 there is the attribute list but not the data characterization. A first question that**

**could have the reader is "Did they authors select one number for each attribute and for each catchment?" or a time series?**

Thank you for raising this point. We will improve Section 3 and include additional information about data characterization to enhance clarity. We will also mention that we do not use the time series data in our analysis. Each catchment in the dataset has five categories of attributes that are outlined in Table 1. We will use this information to categorize catchments (using cluster analysis) based on each category. Therefore, each catchment has been assigned five cluster IDs.

The changes will be as follows:

**From line 219:**

[revised manuscript text omitted]

5. **Figure 1 is not fully clear, Is the cluster analysis necessary? Is it an alternative way to analyze the entire data set? If yes it should be in a different level, like a starting option in the flow chart.**

Regarding Figure 1, we prefer to use Figure 1 to highlight the core of the analysis, which presents the description of the lists of steps of the proposed comparative analysis on causal models and catchment runoff signatures, rather than a direct flowchart. In this figure, cluster analysis uses the entire dataset and assigns cluster IDs to each catchment. We will modify this figure to make it more understandable as follows:

[Figure]

Figure 1: *Flowchart depicting the steps followed in this study. Grey boxes indicate the procedures, orange boxes present the results of these procedures, blue text highlights where information about causality is utilized , and the red text highlights the cluster analysis and indicates where the clustering results are applied.. PC refers to Peter and Clark's causal discovery algorithm, PAM stands for Partition Around the Centroid clustering algorithm, and DAG refers to Directed Acyclic Graph. BN refers to the Bayesian Network, GAM refers to the Generalized Additive Model, and RF refers to the Random Forest. GAMPar and RFPar are causal models (GAM and RF) using only causal parent variables for prediction, while GAMAll and RFAll are non-causal models that use all selected variables as predictors. Baseline models refer to models that use the entire dataset (all 671 catchments) for training and testing, while sub-models use only subsets of the dataset or clusters.*

Regarding the necessity of clustering, When using the entire dataset for training and testing the models, we work with a large, diverse dataset containing a variety of characteristics. However, the question arises: what happens when we focus on regions with fewer catchments that share similar attributes, such as climate, geology, vegetation, topography, or soil properties? To investigate this, we applied clustering analysis to group catchments with similar characteristics, creating categories of varying sizes.

This approach enables us to evaluate the performance of causal and non-causal models across different environments, identifying the situations where the models perform well and where they face challenges. For instance, if the models perform well in a cluster within the topographic category, we can infer that models are effective in regions with specific topographic characteristics and less sensitive to the sample size. Additionally, using causal parents as predictors establishes a causal mechanism for the runoff signature under the assumption that this mechanism remains consistent across different environments. Clustering helps test this assumption and assess the effectiveness of the PC algorithm in identifying the appropriate causal parents using the models' performance. In conclusion, clustering is a complementary rather than an alternative analysis to investigate the causal models further.

6. **The Section 2.2.1 seems incomplete and refers to the Supplementary materials, how-**

**ever this step seems important in the whole procedure. More details on how the most important feature are ranked are necessary, indeed the "out-of-bag method" is vague and the sentence "variables are selected based on a combination of correlation analysis, variable importance assessment and consideration of the underlying physics of the runoff signatures." is too general.**

We agree that the information provided in this section is not sufficient. Therefore, we have added more details to the feature selection section. We will explain that the out-of-bag method is used as a complementary method to the correlation analysis, which helps identify explanatory variables in categories with low correlation coefficients.

The changes will be as follows:

**From line 145:**

(a) *Correlation analysis: Pearson, Kendall, and Spearman correlation coefficients are computed to illustrate the potential explanatory variables. The correlation analysis reveals the most influential variables from each category, namely climate, geology, vegetation, topography and soil. In addition, the scatter plot of the data helped visually understand the relationship between variables.*

(b) *Variable importance: Since the results of the correlation analysis are not always consistent, another feature selection procedure is conducted using the random forest method to investigate the feature importance.  The variables are ranked using  the out-of-bag method, which is quantified using the Mean Decreased Accuracy (IncMSE) score. The out-of-bag method ranks variables based on the increase in prediction error caused by removing each variable from the prediction process. Random forest is implemented using the R package randomForest (Liaw et al., 2015).*

*With the information provided by the procedures mentioned above, variables are selected based on a combination of correlation analysis, variable importance assessment and consideration of the underlying physics of the runoff signatures. We tried to select the most influential variables from each category, including climate, geology, soil, topography, and vegetation. The number of selected variables varies across categories. Multiple variables are selected from categories where most variables exhibit high correlation. Conversely, only the highest-correlated variable is chosen for categories with a weak correlation to the runoff signature of interest. For example, climatic variables are often highly influential for runoff signatures, leading to the selection of multiple variables. In contrast, geological variables tend to show a weak correlation with some runoff signatures, so only the most influential variable from this category is selected. The results of feature selection are presented in the supplementary materials.*

---

## Author Comment (AC2)

**Reply to Reviewer 2**

Thank you very much for taking the time to review our manuscript and for your deep understanding of this topic by providing constructive feedback. We have responded to all of your points and will incorporate the necessary changes into the revised manuscript.

**1 Response to the reviewer's comments**

Please find our responses to your points below:

1. **The authors write that they assume that the runoff signatures are sink nodes ("We also assumed that runoff signatures do not cause climate and catchment attributes") and that there are no hidden variables ("It is also assumed that there are no unobserved variables" – see also the assumption that the distribution is Markov and faithful wrt a DAG over the observed variables). But then, identifying the set of causal parents of a runoff signature becomes 'classical' variable selection, a non-causal problem. This implies that a causal analysis is not necessary.**

   Thank you for pointing out this important aspect. Although identifying the causal parents of a runoff signature might seem similar to a variable selection process, our analysis emphasizes investigating the causal mechanisms that drive the runoff signatures. A variable $Y$ and its causal parents $PA(Y)$ define a causal mechanism where $Y$ is conditionally independent of other variables given its parents. This implies that the parents provide a complete explanation for $Y$ regardless of other covariates. With this understanding, we sought to investigate how these causal parents influence the performance of predictive models and to what extent they can explain the target node. If we were only considering the direct causal parents ($\sim$Par) scenario, in principle, it could be considered a variable selection exercise. However, we are also considering and comparing to the full ($\sim$All) DAG - as well as comparing GAM and RF with and without causality considerations.

   We acknowledge that removing variables during the variable selection step may potentially violate the assumptions we made, which is not uncommon when applying the causal inference method to real-world data. However, applying variable selection before running the causal discovery algorithm is crucial in this context. Including all 41 climate and catchment attributes in the PC algorithm increases the dimensionality of the PC algorithm. Increasing the number of covariates in the PC algorithm can reduce the algorithm's detection power (Runge, 2018). This is because higher dimensionality increases the number of conditional independence tests, which can be unreliable in cases of limited sample size or noisy data (Kalisch and Bühlman, 2007; Li and Wang, 2009; Ramsey et al., 2012). Careful variable selection or dimensionality reduction is often necessary to mitigate these issues (Runge et al., 2023).

   Moreover, we attempted to include all continuous variables during the causal discovery process to account for the possibility of causal sufficiency. Despite this, we observed challenges such as the generation of disconnected DAGs with independent nodes or groups of nodes lacking causal relationships with runoff signatures. We also encountered undirected edges in the graphs, indicating potential hidden confounding. These issues further emphasize the limitations of the PC method in real-world applications. It is important to note that our study represents the first attempt to apply causal discovery algorithms in this specific hydrological context, where no established guidelines exist. This required extensive experimentation, including bootstrapping DAGs, refining data processing methods, and testing various conditional independence tests to derive physically meaningful DAGs.

   Causal discovery methods inherently rely on assumptions, such as those concerning hidden variables in the PC algorithm, which are often violated in real-world problems. In our study, which spans numerous catchments across a broad spatial extent, some degree of approximation was inevitable.

However, future research could explore how these assumptions can be relaxed or adjusted without compromising the validity of the results at different scales. We appreciate your feedback, and these points will be better articulated in the manuscript.

We will incorporate these clarifications into Sections 2.2.1 and 2.2.3 to provide a more comprehensive context for our use of causal discovery methods.

2. **This also has some implications on one of the paper's main points regarding the invariance property of the causal parents of Y: If the environments can be modeled as the values of a random variable (if I understand correctly, the environments are created by clustering certain covariates, so we can indeed model them as a child of such covariate(s)), then not only the set of causal parents but also the full set of covariates is invariant – invalidating some of the main points of the paper.**

This is a good remark that was raised. It is connected to the readability of the manuscript. We do not claim the invariance property of causal parents but the properties of runoff signature ($Y$) given its causal parents ($PA(Y)$). The prediction of a given response is causally dependent on some (selected) covariates; these covariates constitute/can be seen as descriptors of the environment 'driving' the response. Clustering was used to group the CAMELS catchments into different categories based on specific attributes. Any given catchment will belong to one climate attribute cluster, one soil attribute cluster, one topographic attribute cluster, one geological cluster and one vegetation cluster (i.e. each catchment is 'assigned' 5 cluster values, one for each attribute). The whole process of training and testing the models is now (also) done on separate attribute clusters only, so basically, it is only done on a subset of the available data but using data that share certain characteristics. However, the causal parents/selected variables are the same whether we use clustering or not. The covariates in a DAG include variables from all five categories, and in no environment does a covariate serve as a child of a covariate from another environment.

We will improve the readability of Section 2.2.2 by clarifying the distinction between clustering and the use of covariates.

3. **(As a side, empirical differences between 'causal' and 'non-causal' methods are then, in my view, 'only' due to different ways of performing variable selection – and worse performance of non-causal methods on test data simply means that the variable selection or regularization can be improved.)**

Thank you for raising this point. We agree that this point is misleading in our paper, and more explanations need to be added to it. To answer this comment, the variable selection was performed for each runoff signature, and we selected the most influential variables among 41 climate and catchment attributes. The selected variables are used to predict the runoff signature. We called these models non-causal models since they did not use any causal information. Under certain assumptions, we derived the causal relationship between the selected variables. Since, theoretically, the causal parents can explain their child node independent of other covariates, we tested whether using only causal parents affects the prediction model robustness. While this procedure might be similar to variable selection, the main purpose here is not to increase the predictions' accuracy by selecting different sets of predictors but to answer to what extent causal parents can explain their child node compared to using all covariates for prediction. To evaluate this, we used the whole dataset for the prediction, which we called baseline models, and subsets of the dataset, which were called sub-models, with and without utilising causal information. If the causal models performed comparably to or better than non-causal models across different environments, it indicates that causal parents suffice to explain the target variable. In cases where causal models outperformed non-causal ones, it suggests that some covariates in the non-causal models may represent spurious correlations, negatively impacting performance in that specific environment. Overall, the consistency of causal model results highlights the importance of incorporating causal information into prediction frameworks.

We will add these explanations to Section 2.2.

4. **Even if the above three points were not an issue, the authors do not provide sufficient arguments on why we should trust that the result obtained by PC reflects the causal ground truth. A few points why in my view this is not obvious:**

- **The authors use "expert knowledge (…) to determine the causal direction between two variables with an undirected edge, correct the causally wrong direction between variables and block the spurious edges between variables". But if we know that some of the edges are incorrect, why should we trust the others? (Also, I did not find the description of the process of correcting edges sufficiently clear.)**

  We agree that it is necessary to add further explanation of how expert knowledge is used in developing DAGs. Thank you for letting us know. The way we used expert knowledge in our analysis was specifically to determine the causal direction of undirected edges by considering the underlying physics of the processes. Based on this understanding, first, we added impossible links to a blacklist. We then applied the PC algorithm iteratively, correcting the undirected edges and adding them to a whitelist while blacklisting any spurious links, if identified. This procedure continued until the resulting DAG contained no undirected or spurious links. It is worth mentioning that blacklisting impossible links could reduce the number of iterations to reach a 'stable' DAG.

  It is important to note that we do not claim the resulting DAG to represent the ground-truth DAG. If the ground-truth DAG were known, we could evaluate the resulting DAGs using metrics such as the Bayesian Information Criterion (BIC) or Structural Hamming Distance (SHD). However, in the absence of a known ground-truth DAG, the primary means of evaluation relies on domain knowledge. The structure of the DAG can vary depending on the causal discovery method used and the choice of conditional independence tests. The legitimacy of the graph was assessed using our expert knowledge to judge the plausibility of the inferred causal relationships. We also acknowledge that there are objective methods to evaluate the structure of DAGs. For instance, one approach is to generate synthetic data from the obtained DAG, reapply the causal discovery method to the generated data, and then compare the inferred DAG with the original. This validation procedure is among the considerations we plan to explore in future work.

  To address the reviewer's concerns, we will add the following clarifications to the manuscript in Section 2.2.3:

  - An explanation of how expert knowledge was used to correct undirected and spurious edges.
  - A discussion of the limitations of this approach and its implications for the reliability of the resulting DAG.

  We believe these additions will clarify our methodology and address the concerns raised.

- **The PC algorithm is known to produce results that are not reliable. E.g., relabeling the variables, i.e., simply permuting the columns in the data matrix, or subsampling the data set sometimes change the outcome. Simulation experiments show that even under no model misspecification huge sample sizes can be needed to reliably obtain the ground truth graph.**

  Thank you for raising this important point. Indeed, it is well-documented that the PC algorithm's results can be sensitive to factors such as subsampling the dataset or permuting variable order e.g. (Colombo et al., 2014; Kalisch and Bühlman, 2007). To address these limitations in our study, we implemented a structured approach to mitigate the instability of the algorithm's output.

  Firstly, we utilized the entire dataset (671 catchments) to ensure the largest possible sample size for the causal discovery process. Secondly, we iteratively refined the blacklist and whitelist of edges by running the PC algorithm multiple times. This iterative process allowed us to identify spurious, undirected, or unstable edges, which were then systematically added to the respective lists. By doing so, we ensured that the final execution of the PC algorithm, incorporating the blacklist and whitelist, consistently produced the same DAG.

  While this was a labour-intensive and largely manual task, it represents a step toward addressing the algorithm's inherent variability. We also acknowledge that this process could be further improved and potentially automated in future studies.

  This discussion will be added to section 2.2.3.

- **The paper does not provide any theoretical guarantees. This would probably be too much to ask for an applied paper but I argue below that this question is**

**not purely theoretical: the assumptions that are known to be sufficient to obtain theoretical guarantees are most likely violated in this application.**

We acknowledge that this study focuses on applying existing methods rather than exploring their theoretical foundations or the assumptions underlying them. However, introducing structural changes to the DAGs across different environments would indeed require theoretical guarantees to ensure that the assumptions used to construct DAGs for the baseline models remain valid for the sub-models with modified DAGs.

In this study, the structure of the DAGs and their variables remains consistent across all environments for each runoff signature. As a result, the assumption of causal sufficiency and the independence of the runoff signature from other covariates, given its parent nodes, holds across all environments. It should be noted that the causal sufficiency assumption is often violated in real-world applications. According to Runge (2018), to satisfy causal sufficiency, we only need to assume that no unobserved variables directly or indirectly influence any pair of our measured variables when working with a limited set of observed data. While we acknowledge that the assumptions underlying causal discovery methods may be strong, combining well-established methods, careful preprocessing, and domain expertise helps mitigate the impact of potential assumption violations.

5. **The paper does not provide sufficient arguments on whether the differences between methods are statistically significant.**

Thank you for pointing this out. We conducted a random sampling of the training and testing sets 100 times, enabling us to shuffle the catchments between these sets. This approach allowed us to simulate runoff signatures using various combinations of catchment attributes within each environment and to compare the averaged R-squared and RMSE values for each model.

To address this comment, we performed a nested F-test to calculate the statistical significance of differences between causal ($\sim$Par) and non-causal ($\sim$All) GAM models for train and test results. For random forest, which is a non-parametric and non-linear model, we performed a permutation test to calculate the statistical significance of causal ($\sim$Par) and non-causal ($\sim$All) RF models for training and test results. The details and the results of the tests are presented in the next section. The results of this analysis will be added to the supplementary materials. The tests' results suggest that although the difference between causal and non-causal models is significant during the training phase, it often becomes insignificant during testing.

6. **In my view, the paper is not sufficiently clear about the experimental setup using the different clusters. E.g., how exactly are the training and test sets chosen? The authors mention robustness across environments but then training and test data should be from different clusters?**

Thank you for letting us know about this issue. In this study, the environments correspond to clusters in the sub-models and the entire dataset in the baseline models. For each environment, we divide the dataset into training and test sets, where 75% of the catchments are randomly selected for training, and the remaining 25% are used for testing. The process of selecting catchments for the training and testing sets is repeated iteratively 100 times, creating different combinations of catchments in these sets for each environment. This approach provides a range of model performances, and their average performance is used for comparison. Importantly, training and testing are conducted within the same environment. Therefore, if a model is trained on catchments from a specific cluster, for example, climate cluster 1, it is also tested on catchments within that same cluster. We will add more explanations on the experimental setup at the end of Section 2.2.

7. **It was unclear to me how the paper accounts for time-dependence of the data points.**

In this work, we did not use time-dependent variables. Climatic indices and runoff signatures are derived from their respective time series for the whole period from 01/10/1989 to 30/09/2009. These values represent time-aggregated properties of the time series. Additionally, variables related to attributes such as topography, soil, vegetation, and geology can be considered time-independent. The obtained DAGs in this study are static Bayesian Networks.

If the question is about time dependence in the causal order, time-averaging does not eliminate the underlying causal mechanisms (Gong et al., 2017). The relationships identified in the static DAG still hold because they reflect aggregated causal effects that persist over time. The causal ordering

thus remains valid in this aggregated representation. We will add an explanation to Section 2.2 to clarify this point.

8. **(As a side, in general, when considering robustness against a change of environment, using the causal parents as covariates may not be optimal. Instead, one could use what is referred to as the stable blanket.)**

Thank you for this comment. In our study, the causal structure (DAG) remains consistent, with the target variables (runoff signatures) having no child nodes, and the set of causal parents remaining unchanged across all environments. As a result, the causal parents align with both the Markov blanket and the stable blanket. This is because the causal parents of the target variable form a subset of the Markov blanket, and interventions on non-parent nodes do not alter the functional relationships governing the causal mechanism of the target variable (Pfister et al., 2021). Therefore, in our case, we can consider the causal parents as the stable blanket.

9. **The paper contains several imprecise/incorrect statements. Here are two examples (there are more): "They are the assumptions under which the causal relationship from the observational data can be learned." What precisely does "can be learned" mean? This may sound like a minor point but in my view it is not. One way of making this precise is to write down conditions for uniform consistency. There are few conditions known under which uniform consistency holds. However, such conditions are very restrictive. (E.g., some of such conditions include the assumption that the random variables are jointly Gaussian. If I understand correctly, the authors transform marginals but even this does not suffice.) It is known that, in general, all nonparametric conditional independence tests that are level are trivial (and do not have any non-trivial power), so it may even be impossible to relax such conditions to something reasonable. This is important in that these thoughts may be a reason for why the PC algorithm is usually unreliable in practice (see above). To give another example, "Covariate shift states that if variable Y is to be predicted from X, and X is the cause of Y, the conditional probability P(Y—X) remains the same across all environments if the distribution of X changes" is in my view at least imprecise: covariate shift is usually meant as a non-causal assumption and invariance generally holds only if X is the set of all causal parents of Y.**

Thank you for your detailed feedback.

- On the phrase "can be learned": By "can be learned," we mean "can be discovered," as these terms are often used interchangeably in the causal inference literature. Common terminologies include "causal learning," "causal discovery", and "learning causal structure" (Peters et al., 2017). To avoid ambiguity, we will standardize these terms throughout the paper to ensure clarity and consistency.

- On the assumptions for uniform consistency and Gaussian transformation: Gaussianity is one of the assumptions of PC algorithm. Regarding the data transformation to approximate a Gaussian distribution, we followed the approach outlined in the literature. For example, in (Dutta and Maity, 2020), a similar transformation process was employed. We acknowledge that Gaussian assumptions can be restrictive and do not address all the limitations you've raised. However, we adopted this approach as it aligns with established practices in the field and enables the use of conditional independence tests that assume Gaussianity.

- On the description of covariate shift: As you correctly noted, covariate shift is a non-causal assumption. Additionally, the invariance of $P(Y \mid X)$ indeed holds when $X$ includes all causal parents of $Y$. In our work, we did not claim otherwise. Specifically, we ensured that $X$ contains all causal parents of $Y$ as relevant to our analysis. The environments in our study are constructed using clustering, which creates subsets of catchments containing the same variables but with differing distributions. These environments are designed to test the causal mechanisms (runoff signatures and their causal parents) under varying conditions inspired by the principles of covariate shift.

We will revise the text to reflect these points clearly and accurately to avoid any potential misunderstandings.

10. **The paper contains several typos, such as "casual" or "Clarck" or "causal models are assum result".**

   We will fix this issue in the entire manuscript.

**2 Changes in the manuscript**

In this section, the main changes that will be made in the manuscript is shown:

**2.2 Methods**

The methodology integrates feature selection, clustering, causal discovery and prediction. Fig. 1 shows the methodological procedure used in this study. In Fig. 1, causal models refer to the models that use causal parents, and non-causal models use all selected variables as predictors. Environments are subsets of the dataset obtained by clustering algorithms. Therefore, the words environment, cluster and subset imply the same meaning in this study. Baseline models refer to the models that use the whole dataset, all 671 catchments, for training and testing, and sub-models use subsets of the dataset for this purpose. GAM~Par and RF~Par are causal GAM and RF models that employ causal parents for prediction. GAM~All and RF~All are non-causal GAM and RF models that use all the selected variables as predictors. A robust model is defined as one that maintains its accuracy across different environments.

Since, theoretically, the causal parents contain all information to explain their child node independent of other covariates, we tested whether using only causal parents affects the prediction model robustness.  The primary goal here is not merely to improve prediction accuracy by selecting different sets of predictors. Instead, it is to assess how well causal parents can explain their child node compared to using all covariates for prediction and to determine how integrating causal information can enhance the parsimony and robustness of prediction models.

To evaluate this, we used the whole dataset for the prediction (baseline models) and subsets of the dataset (sub-models) with and without utilising causal information and causal and non-causal models, respectively. If the causal models performed comparably to or better than non-causal models across different environments, it indicates that causal parents suffice to explain the target variable. In cases where causal models outperformed non-causal ones, it suggests that some covariates in the non-causal models may represent spurious correlations, negatively impacting performance in that specific environment. Furthermore, the robustness of the models is assessed by comparing their accuracy in training and test settings.

The steps are explained in the following sections.

**Section 2.2; from line 214:**
For all models, BN, GAM, and RF,  and for each environment, we divide the dataset into training and test sets, where 75% of the catchments are randomly selected for training, and the remaining 25% are used for testing. This process is repeated 100 times using bootstrapping to generate different combinations of training and test sets. This approach provides a range of model performances, and their average performance is used for comparison. Importantly, training and testing are conducted within the same environment. For example, if a model is trained on catchments from a specific climate category cluster, it is also tested on catchments within that same cluster. The models are executed for the whole dataset (baseline models) and each cluster of categories (sub-models). The models' accuracy is evaluated using Root Mean Squared Error (RMSE) and R-squared metrics between prediction and observations. The iteration provides 100 RMSE and R-squared for each run, and the accuracy is reported as their mean value. The following section discusses the obtained results of this study.

**2.2.1 Feature Selection**
In this section, we conduct the variables selection to 1) identify the most influential factors explaining the target signature and 2) reduce the dimensionality of the causal discovery problem (Runge et al., 2023). Including all 41 climate and catchment attributes in the PC algorithm increases its dimensionality, which can have adverse effects. A higher number of covariates reduces the statistical significance of detected edges and increases the risk of spurious links. This occurs because high dimensionality requires more

conditional independence tests, which can become unreliable in cases of limited sample sizes (Kalisch and Bühlman, 2007; Li and Wang, 2009; Le et al., 2016; Ramsey et al., 2012). It is worth mentioning that we attempted to include all continuous variables in the causal discovery process without applying variable selection. This approach was tested to address the causal sufficiency assumption in the PC algorithm. Despite this, we observed challenges such as the generation of disconnected DAGs with independent nodes or groups of nodes lacking causal relationships with runoff signatures.

The explanatory variables for each signature are selected based on 1) ranked correlation coefficients and 2) variable importance. It should be noted that to develop the BN, which is a probabilistic graphical model, the selected variables (nodes) shouldn't be the deterministic functions of each other; otherwise, the conditional dependency structure of DAGs will change. Therefore, the aridity index, a function of precipitation and potential evapotranspiration, is removed from the selection procedures. Additionally, it is assumed that the selected variables satisfy causal Markov and faithfulness assumptions (Spirtes et al., 2001) when used for the PC causal discovery algorithm. They are the assumptions under which the causal relationship from the observational data can be learned. These assumptions relate the d-separation in the graph to conditional dependencies in the joint distribution (Pearl, 2009). These assumptions are explained in the following sections. The methods used for correlation analysis and variable importance are as follows:

[revised manuscript text omitted]

We do not claim the resulting DAG to be the ground-truth DAG using this procedure. If the ground-truth or reference DAGs were known, we could evaluate the resulting DAGs using metrics such as the Bayesian Information Criterion (BIC) or Structural Hamming Distance (SHD). However, in the absence of a known ground-truth DAG, the primary means of evaluation relies on domain knowledge. The structure of the DAG can vary depending on the causal discovery method used and the choice of conditional independence tests. The legitimacy of the graph was assessed using our expert knowledge to judge the plausibility of the inferred causal relationships.

The obtained DAG structures are used to predict runoff signatures using Bayesian Network methods. Additionally, Generalized Additive Models and Random Forests are applied to predict runoff signatures: once using all variables in the DAGs (non-causal models) and once using only the causal parents of the target nodes (causal models). Since the target variable (runoff signature) has no child nodes, its causal parents provide an optimal blanket for the regression models. This is because the causal parents

form a subset of the Markov blanket, and interventions on non-parent nodes do not affect the functional relationships underlying the causal mechanism of the target variable (Pfister et al., 2021).

**Supplementary**

**12 Significance of Differences Between Causal and Non-Causal Models**

**12.1 Difference between GAM∼Par and GAM∼All**

To determine whether the differences between causal and non-causal GAM models are statistically significant, we perform a nested F-test. This test is chosen because the causal model is nested within the non-causal model. We did this test to compare the train and test results of the causal (GAM∼Par) and non-causal (GAM∼All) models. If the result of this test is not significant (P-value >0.05) it means that GAM∼All is not significantly better than GAM∼Par. The steps to perform the nested F test are as follows:

1. Hypotheses:

    - $H_0$: The simpler (nested/causal) model provides an adequate fit to the data.
    - $H_1$: The more complex model (non-causal) provides a significantly better fit.

2. F-statistic: The F-statistic is calculated as:

$$F = \frac{\frac{\text{RSS}_{\text{nested}} - \text{RSS}_{\text{full}}}{df_{\text{full}} - df_{\text{nested}}}}{\frac{\text{RSS}_{\text{full}}}{n - df_{\text{full}}}}$$

    where:

    - $\text{RSS}_{\text{nested}}$: Residual sum of squares of the nested (simpler/causal) model
    - $\text{RSS}_{\text{full}}$: Residual sum of squares of the full (complex/non-causal) model
    - $df_{\text{nested}}$: Residual degree of freedom of the nested model
    - $df_{\text{full}}$: Residual degree of freedom the full model
    - $n$: Number of observations

3. Derive P-value: The F-statistic follows an $F$-distribution with degrees of freedom:

$$\text{df}_1 = df_{\text{full}} - df_{\text{nested}}, \quad \text{df}_2 = df_{\text{full}}$$

4. Decision Rule: Compare the calculated $F$-statistic to the critical value from the $F(\text{df}_1, \text{df}_2)$-distribution or use the corresponding P-value. Reject $H_0$ if:

$$P < \alpha$$

    where $\alpha$ is the significance level and is equal to 0.05.

Table 1: Statistical significance of differences between causal (GAM∼Par) and non-causal (GAM∼All) models. The significance level ($a$) is set to 0.05, and P-values are calculated using the nested F-test for both train and test results. The stars, **\***, indicate statistically significant differences between causal and non-causal models, and **NS** stands for Not Significant within the respective environment, indicating that the null hypothesis has not been rejected. In the "Environment" column, "Clim" refers to climate, "Geol" to geology, "Topo" to topography, and "Vege" to vegetation.

**Statistical significance of difference between GAM∼Par and GAM∼All**

| Environment | Baseflow Index Train | Baseflow Index Test | High Q Dur Train | High Q Dur Test | High Q Freq Train | High Q Freq Test | Low Q Dur Train | Low Q Dur Test | Low Q Freq Train | Low Q Freq Test | Q mean Train | Q mean Test |
|---|---|---|---|---|---|---|---|---|---|---|---|---|
| Baseline | * | NS | * | NS | * | NS | * | NS | * | NS | * | NS |
| Clim 1 | * | NS | * | NS | * | NS | * | NS | * | NS | * | * |
| Clim 2 | * | NS | * | NS | * | NS | * | NS | * | NS | * | NS |
| Clim 3 | * | NS | * | NS | * | NS | * | NS | * | NS | * | NS |
| Clim 4 | NS | NS | * | NS | * | NS | * | NS | * | NS | * | NS |
| Geol 1 | * | NS | * | NS | * | NS | * | NS | * | NS | * | NS |
| Geol 2 | NS | NS | * | NS | * | NS | * | NS | * | NS | * | NS |
| Geol 3 | * | NS | * | NS | * | NS | * | NS | * | NS | * | NS |
| Geol 4 | * | NS | * | NS | * | NS | * | NS | * | NS | * | NS |
| Geol 5 | * | NS | * | NS | * | NS | * | NS | * | NS | * | NS |
| Geol 6 | * | NS | * | NS | * | NS | * | NS | * | NS | * | NS |
| Geol 7 | * | NS | NS | NS | NS | NS | NS | NS | NS | NS | NS | NS |
| Soil 1 | * | NS | * | NS | * | NS | * | NS | * | NS | * | NS |
| Soil 2 | * | NS | * | NS | * | NS | * | NS | * | NS | * | NS |
| Soil 3 | * | NS | * | NS | * | NS | * | NS | * | NS | * | NS |
| Soil 4 | * | NS | * | NS | * | NS | * | NS | * | NS | * | NS |
| Soil 5 | * | NS | * | NS | * | NS | * | NS | * | NS | * | NS |
| Soil 6 | * | NS | * | NS | * | NS | * | NS | * | NS | * | NS |
| Topo 1 | * | NS | * | NS | * | NS | * | NS | * | NS | * | NS |
| Topo 2 | * | NS | * | NS | * | NS | * | NS | * | NS | * | NS |
| Topo 3 | * | NS | * | NS | * | NS | * | NS | * | NS | * | NS |
| Topo 4 | * | NS | * | NS | * | NS | * | NS | * | NS | * | NS |
| Vege 1 | * | NS | * | NS | * | NS | * | NS | * | NS | * | NS |
| Vege 2 | * | NS | * | NS | * | NS | * | NS | * | NS | * | NS |
| Vege 3 | * | NS | * | NS | * | NS | * | NS | * | NS | * | NS |
| Vege 4 | * | NS | * | NS | * | NS | * | NS | * | NS | * | NS |
| Vege 5 | * | NS | * | NS | * | NS | * | NS | * | NS | * | NS |
| Vege 6 | * | NS | * | NS | * | NS | * | NS | * | NS | * | NS |

Table 1: (continued) Statistical significance of differences between causal (GAM∼Par) and non-causal (GAM∼All) models. The significance level ($a$) is set to 0.05, and P-values are calculated using the nested F-test for both train and test results. The stars, **\***, indicate statistically significant differences between causal and non-causal models, and **NS** stands for Not Significant within the respective environment, indicating that the null hypothesis has not been rejected. In the "Environment" column, "Clim" refers to climate, "Geol" to geology, "Topo" to topography, and "Vege" to vegetation.

**Statistical significance of difference between GAM∼Par and GAM∼All**

| Environment | Q5 Train | Q5 Test | Q95 Train | Q95 Test | Runoff Ratio Train | Runoff Ratio Test | Slope of FDC Train | Slope of FDC Test | Stream Elas Train | Stream Elas Test |
|---|---|---|---|---|---|---|---|---|---|---|
| Baseline | * | NS | * | NS | * | * | * | NS | * | NS |
| Clim 1 | * | NS | * | * | * | * | * | NS | * | NS |
| Clim 2 | * | NS | * | NS | * | NS | * | NS | * | NS |
| Clim 3 | * | NS | * | NS | * | NS | * | NS | * | NS |
| Clim 4 | * | NS | * | NS | * | NS | * | NS | * | NS |
| Geol 1 | * | NS | * | NS | * | NS | * | NS | * | NS |
| Geol 2 | * | NS | * | NS | * | NS | * | NS | * | NS |
| Geol 3 | * | NS | * | NS | * | NS | * | NS | * | NS |
| Geol 4 | * | NS | * | NS | * | NS | * | NS | * | NS |
| Geol 5 | * | NS | * | NS | * | NS | * | NS | * | NS |
| Geol 6 | * | NS | * | NS | * | NS | * | NS | * | NS |
| Geol 7 | NS | NS | NS | NS | NS | NS | NS | NS | NS | NS |
| Soil 1 | * | NS | * | NS | * | NS | * | NS | * | NS |
| Soil 2 | * | NS | * | NS | * | NS | * | NS | * | NS |
| Soil 3 | NS | NS | * | NS | * | NS | * | NS | * | NS |
| Soil 4 | NS | NS | * | NS | * | NS | * | NS | * | NS |
| Soil 5 | * | NS | * | NS | * | NS | * | NS | * | NS |
| Soil 6 | * | NS | * | NS | * | NS | * | NS | * | NS |
| Topo 1 | * | NS | * | NS | * | NS | * | NS | NS | NS |
| Topo 2 | * | NS | * | NS | * | NS | * | NS | * | NS |
| Topo 3 | * | NS | * | NS | * | NS | * | NS | * | NS |
| Topo 4 | * | NS | * | NS | * | NS | * | NS | * | NS |
| Vege 1 | * | NS | * | NS | * | NS | * | NS | NS | NS |
| Vege 2 | * | NS | * | NS | * | NS | * | NS | * | NS |
| Vege 3 | * | NS | * | NS | * | NS | * | NS | * | NS |
| Vege 4 | NS | NS | * | NS | * | NS | * | NS | * | NS |
| Vege 5 | * | NS | * | NS | * | NS | * | NS | * | NS |
| Vege 6 | * | NS | * | NS | * | NS | * | NS | * | NS |

**12.2 Difference between RF∼Par and RF∼All**

To assess the significance of the difference between the causal (∼Par) and non-causal (∼All) random forest (RF) models, we employ a non-parametric permutation test. This method is appropriate because random forests are non-parametric and nonlinear, making methods like the nested F-test unsuitable. In this test, the $R^2$ and $RMSE$ values of training obtained from 100 runs of each model are resampled (with replacement) to construct a null hypothesis distribution of performance differences. This is achieved by randomly shuffling the labels of RF∼All and RF∼Par, recalculating the performance difference for each shuffle. A total of 10,000 permutations are performed to ensure the robustness of the null distribution for both train and test results. The P-value is then determined as the proportion of permuted differences that are as large or larger than the observed difference. A significance threshold ($a$) of 0.05 is used to evaluate the results. If the obtained P-value is greater than 0.05, the difference between RF∼All and RF∼Par is not statistically significant.

Table 2: Statistical significance of differences between causal (RF∼Par) and non-causal (RF∼All) models. The significance level ($a$) is set to 0.05, and P-values are calculated using the permutation test for both train and test results. The stars, *, indicate statistically significant differences between causal and non-causal models, and **NS** stands for Not Significant within the respective environment. In the "Environment" column, "Clim" refers to climate, "Geol" to geology, "Topo" to topography, and "Vege" to vegetation.

Statistical significance of difference between RF∼Par and RF∼All

| Environment | Baseflow Index Train | Baseflow Index Test | High Q Dur Train | High Q Dur Test | High Q Freq Train | High Q Freq Test | Low Q Dur Train | Low Q Dur Test | Low Q Freq Train | Low Q Freq Test | Q mean Train | Q mean Test |
|---|---|---|---|---|---|---|---|---|---|---|---|---|
| Baseline | * | * | * | * | * | * | * | * | * | * | * | * |
| Clim 1 | * | * | * | * | * | * | * | NS | * | * | * | * |
| Clim 2 | * | * | * | * | * | * | * | * | * | * | * | NS |
| Clim 3 | * | NS | * | NS | * | * | * | * | * | * | * | * |
| Clim 4 | * | * | * | * | * | * | * | * | * | * | * | NS |
| Geol 1 | * | NS | * | * | * | * | * | NS | * | * | * | NS |
| Geol 2 | * | * | * | NS | * | * | * | NS | * | NS | * | NS |
| Geol 3 | * | * | * | NS | * | * | * | * | * | * | * | NS |
| Geol 4 | * | * | * | NS | * | * | * | NS | * | * | * | NS |
| Geol 5 | * | * | NS | NS | * | NS | * | * | * | * | * | NS |
| Geol 6 | * | NS | * | NS | * | NS | * | NS | * | NS | NS | * |
| Geol 7 | * | NS | * | * | * | NS | * | NS | * | NS | * | NS |
| Soil 1 | * | NS | * | * | * | * | * | NS | * | NS | * | NS |
| Soil 2 | * | * | * | NS | * | * | * | NS | * | * | NS | NS |
| Soil 3 | * | NS | * | * | * | * | * | NS | * | NS | * | NS |
| Soil 4 | * | NS | * | * | * | * | * | * | * | NS | * | NS |
| Soil 5 | * | * | * | NS | * | * | * | NS | * | * | * | NS |
| Soil 6 | * | * | * | NS | * | NS | * | NS | * | * | NS | NS |
| Topo 1 | * | * | * | NS | * | * | * | * | * | * | NS | NS |
| Topo 2 | * | * | * | NS | * | * | * | NS | * | * | * | NS |
| Topo 3 | * | * | * | NS | * | * | * | * | * | * | * | * |
| Topo 4 | * | NS | * | * | * | * | * | * | * |  | * | NS |
| Vege 1 | * | * | * | NS | * | * | * | * | * | * | * | NS |
| Vege 2 | * | * | * | NS | * | * | * | NS | * | * | * | * |
| Vege 3 | * | * | * | NS | * | * | * | NS | * | * | * | NS |
| Vege 4 | * | NS | * | NS | * | NS | * | NS | * | NS | * | NS |
| Vege 5 | * | * | * | NS | * | * | * | * | * | * | * | NS |
| Vege 6 | * | NS | * | * | * | * | * | * | * | * | * | NS |

Table 2: (continued) Statistical significance of differences between causal (RF~Par) and non-causal (RF~All) models. The significance level ($a$) is set to 0.05, and P-values are calculated using the permutation test for both train and test results. The stars, **\***, indicate statistically significant differences between causal and non-causal models, and **NS** stands for Not Significant within the respective environment. In the "Environment" column, "Clim" refers to climate, "Geol" to geology, "Topo" to topography, and "Vege" to vegetation.

**Statistical significance of difference between RF~Par and RF~All**

| Environment | Q5 Train | Q5 Test | Q95 Train | Q95 Test | Runoff Ratio Train | Runoff Ratio Test | Slope of FDC Train | Slope of FDC Test | Stream Elas Train | Stream Elas Test |
|---|---|---|---|---|---|---|---|---|---|---|
| Baseline | * | * | * | * | * | * | * | NS | * | NS |
| Clim 1 | * | * | * | * | * | * | * | * | * | * |
| Clim 2 | * | NS | * | NS | * | * | * | * | * | * |
| Clim 3 | * | NS | * | NS | * | * | * | NS | * | * |
| Clim 4 | * | NS | * | NS | * | * | * | * | * | NS |
| Geol 1 | NS | NS | * | NS | * | * | * | NS | * | NS |
| Geol 2 | NS | NS | * | NS | * | NS | * | * | * | * |
| Geol 3 | * | NS | * | NS | * | * | * | NS | * | NS |
| Geol 4 | NS | NS | * | NS | * | * | * | NS | * | NS |
| Geol 5 | * | * | * | NS | * | NS | * | * | * | * |
| Geol 6 | * | * | * | NS | * | * | * | NS | * | NS |
| Geol 7 | * | NS | * | NS | * | * | * | NS | * | NS |
| Soil 1 | * | NS | * | NS | * | * | * | NS | * | NS |
| Soil 2 | * | NS | * | NS | * | * | NS | NS | * | * |
| Soil 3 | * | NS | * | NS | * | * | * | * | * | * |
| Soil 4 | * | NS | * | NS | * | NS | * | * | * | NS |
| Soil 5 | * | * | * | NS | * | * | * | * | * | NS |
| Soil 6 | NS | NS | * | NS | * | * | * | * | * | NS |
| Topo 1 | * | * | * | NS | * | * | * | * | * | * |
| Topo 2 | * | NS | * | * | * | * | * | * | * | * |
| Topo 3 | * | NS | * | NS | * | NS | * | * | * | * |
| Topo 4 | * | NS | * | NS | * | * | * | * | * | NS |
| Vege 1 | * | * | * | * | * | * | * | * | * | NS |
| Vege 2 | * | * | * | NS | * | * | * | * | * | * |
| Vege 3 | * | * | * | * | * | NS | * | * | * | NS |
| Vege 4 | * | NS | * | NS | * | NS | * | * | * | NS |
| Vege 5 | * | NS | * | * | * | * | * | * | * | NS |
| Vege 6 | * | NS | * | * | * | * | * | * | * | * |

---

## Referee Report (RR1)

**1 Comments on the authors' response:**

1) First of all, I want to point out that the output of the PC algorithm is a CPDAG, i.e., the Markov equivalence class. A CPDAG is a graphical representation of a set of DAGs where the distribution satisfies the Markov property relative to every single DAG in that set. This means that given a distribution which satisfies the Markov property for one DAG, there could also be several other DAGs for which the Markov assumption is satisfied. In other words, given a distribution, there are several DAGs where the distribution fulfills the Markov assumptions with respect to those DAGs. When orienting undirected edges to obtain a DAG from this set of DAGs, you should be careful not to introduce new unshielded colliders / v-structures. Additionally, you can also introduce background knowledge before running the PC algorithm. See for example "Interpreting and using CPDAGs with background knowledge" (2017) or *"Constraint-based causal discovery with tiered background knowledge and latent variables in single or overlapping datasets"* (2025).

2) Depending on the specific algorithm and plot function you are using, you will obtain bi-directed edges. If you use R and the pcalg package, there are cases where the direction could not be determined (but not because of the Markov equivalence class). This leads to an invalid CPDAG, meaning that your output is not representing a Markov equivalence class. There are several reasons for this, but it is important to mention that undirected and bi-directed edges are not the same. There are at least three violations of assumptions which may lead to an invalid CPDAG: cycles, hidden common causes, and selection bias. It could be that you set a direction for an edge without realizing that you are violating something else (for example, introducing new unshielded colliders / v-structures or creating cycles).

3) You wrote: *"It is worth mentioning that we attempted to include all continuous variables in the causal discovery process without applying variable selection. This approach was tested to address the causal sufficiency assumption in the PC algorithm, which requires that all common causes of the target variables are accounted for. Despite this, we observed challenges such as the generation of disconnected DAGs with independent nodes or groups of nodes lacking causal relationships with runoff signatures."*

This should be a big red warning signal for your analysis and needs further investigation. The output of the PC algorithm heavily depends on the alpha value, this is the significance level for the tests. Have you optimized this value somehow? How does the results depends on alpha? Additionally, it seems strange to me to perform feature selection methods before applying the PC algorithm. The PC algorithm also estimates Pearson correlations, and the first step of the PC algorithm (learning the skeleton) is based purely on conditional independence tests. These tests are, for example, partial correlation for two variables conditional on a third one, or in the first step just a correlation between two variables. Why should you use an extra step of correlation analysis if the PC algorithm will do the same and additionally will save the information on potential unshielded colliders / v-structures?

4) Furthermore, the authors are mainly interested in discovering the parent set of the outcome Y. Why are you not considering methods which are designed for this task? I mean Invariant Causal Prediction (ICP). I would spend a bit more time on the limits of causal discovery and non-linear methods. I would recommend reading:

1. Model-Based Causal Feature Selection for General Response Types (2024)

2. Invariant Causal Prediction for Nonlinear Models (2018)

3. Causal inference by using invariant prediction: identification and confidence intervals (2016)

5) Furthermore, you are using GAMs without specifying which GAMs you are using. Which link function do you use? Which family of distribution is assumed for the outcome y? Generalized linear models are then useful if your outcome is discrete (Poisson, ...), binary (logistic, c-log-log, ...), or continuous but only positive valued (exponential, gamma, ...). I see why you are using the additive components (splines), but I miss some more information about the model specification. Without this information, reproducibility is not possible.

6) A further point to note is that while the PC algorithm basically only uses (partial) correlation, testing linear dependency, the subsequent use of additive models assumes a potentially non-linear association

between X and the mean of Y. This is not necessarily a flaw in your approach, but it is crucial to be aware that while the PC algorithm relies on (partial) correlations, which inherently assess linear relationships (keeping in mind the fundamental connection between linear regression and correlation), the application of GAMs (the additive part of it) implies that a non-linear relationship between the predictors and the mean of y is assumed.

What you could try is something like using regression modeling with cubic spline as a heuristic test of conditional independence. See for an data example: "Data-Driven Model Building for Life-Course Epidemiology (2021)".

7) While the paper makes a good effort to explore a new idea in prediction of hydrological responses, I think there are some areas where the authors could show a better awareness of the methods' limitations and assumptions they're using. It's important to acknowledge that this isn't entirely the authors' "fault"; causal discovery research is still developing, and it's not always straightforward to apply these methods to any kind of data. The PC algorithm, for instance, tends to work best with multivariate Gaussian data, perhaps with a few categorical variables, and when you have a large enough sample size and a well-chosen alpha value (probably only given in simulations). Applying it to more complex data with non-linear relationships and arbitrary data types can be very challenging. Therefore, I would not suggest using the PC algorithm, instead trying more task-specific methods, such as ICP mentioned above.

---

## Author Response (AR2)

**Reply to Reviewer 3**

Thank you very much for taking the time to review our manuscript and for providing constructive feedback. Your comments have helped improve the quality of our work. We have revised the manuscript and repeated all analyses, taking into account the points you raised. Additionally, we increased the number of model runs from 100 to 500 using a high-performance computing (HPC) system. The updated code used in our analysis has also been uploaded to GitHub for transparency and reproducibility. We hope that the changes we have made address your concerns. Below, we summarize the main points we have revised:

1. **First of all, I want to point out that the output of the PC algorithm is a CPDAG, i.e., the Markov equivalence class. A CPDAG is a graphical representation of a set of DAGs where the distribution satisfies the Markov property relative to every single DAG in that set. This means that given a distribution which satisfies the Markov property for one DAG, there could also be several other DAGs for which the Markov assumption is satisfied. In other words, given a distribution, there are several DAGs where the distribution fulfills the Markov assumptions with respect to those DAGs. When orienting undirected edges to obtain a DAG from this set of DAGs, you should be careful not to introduce new unshielded colliders / v-structures. Additionally, you can also introduce background knowledge before running the PC algorithm. See for example "Interpreting and using CPDAGs with background knowledge" (2017) or "Constraint-based causal discovery with tiered background knowledge and latent variables in single or overlapping datasets" (2025).**

   Thank you for this comment. We totally agree that we should be aware not to introduce any new unshielded colliders when orienting undirected edges to obtain the DAG. To derive DAGs from CPDAGs, we previously relied on expert knowledge before and after running the PC algorithm. Therefore, to ensure the validity of our DAGs, and in response to your third comment regarding variable selection, we repeated the entire analysis using a consistent set of 22 variables across all runoff signature (target) variables. In this revised analysis, we incorporated general edge assumptions as background knowledge, considering the references you provided, prior to running the PC algorithm. We placed these implausible links in a blacklist and then ran the PC algorithm. This helped have a valid CAPDAG with a minimum number of Markov equivalence classes. Using this blacklist, we obtained a CPDAG with only one undirected edge between mean elevation and mean slope. Since neither of these variables has any parent nodes, the resulting CPDAG corresponds to two Markov equivalence classes. Introducing a directed edge, either from mean elevation to mean slope or vice versa, does not create any unshielded colliders. Therefore, we oriented the edge from mean elevation to mean slope. These points are addressed in the newly added Section 3.2 of the Methods, and we have rewritten and updated the entire explanation of the causal discovery section accordingly.

2. **Depending on the specific algorithm and plot function you are using, you will obtain bi-directed edges. If you use R and the pcalg package, there are cases where the direction could not be determined (but not because of the Markov equivalence class). This leads to an invalid CPDAG, meaning that your output is not representing a Markov equivalence class. There are several reasons for this, but it is important to mention that undirected and bi-directed edges are not the same. There are at least three violations of assumptions which may lead to an invalid CPDAG: cycles, hidden common causes, and selection bias. It could be that you set a direction for an edge without realizing that you are violating something else (for example, introducing new unshielded colliders / v-structures or creating cycles).**

   Thank you for raising this point and for your clear explanation. We used the *bnlearn* package in R along with the *graphviz.plot* function to generate the CPDAGs, which correctly displays undirected

edges when appropriate. To enhance visual clarity and ensure consistency across all figures, we manually recreated the graphs using Inkscape, a vector graphics editor. No bidirected edges were introduced during the causal discovery process. Additionally, we took care not to orient any edges in a way that would violate the assumptions of a valid CPDAG, such as by creating cycles, introducing unshielded colliders (v-structures), or misrepresenting the underlying Markov equivalence class.

3. **You wrote: "It is worth mentioning that we attempted to include all continuous variables in the causal discovery process without applying variable selection. This approach was tested to address the causal sufficiency assumption in the PC algorithm, which requires that all common causes of the target variables are accounted for. Despite this, we observed challenges such as the generation of disconnected DAGs with independent nodes or groups of nodes lacking causal relationships with runoff signatures." This should be a big red warning signal for your analysis and needs further investigation. The output of the PC algorithm heavily depends on the alpha value, this is the significance level for the tests. Have you optimized this value somehow? How does the results depends on alpha? Additionally, it seems strange to me to perform feature selection methods before applying the PC algorithm. The PC algorithm also estimates Pearson correlations, and the first step of the PC algorithm (learning the skeleton) is based purely on conditional independence tests. These tests are, for example, partial correlation for two variables conditional on a third one, or in the first step just a correlation between two variables. Why should you use an extra step of correlation analysis if the PC algorithm will do the same and additionally will save the information on potential unshielded colliders / v-structures?**

Thank you for this comment. We acknowledge that the detected edges by the PC algorithm are sensitive to the choice of the alpha level, which controls the threshold for conditional independence tests. In our initial experiments using all available continuous variables (without feature selection), we applied the alpha value of 0.05. However, this resulted in sparse CPDAGs, sometimes with disconnected nodes, for example, geological permeability. Additionally, domain-reasonable relationships, such as the influence of geological attributes on the baseflow index, failed to appear under this setting. Therefore, applying the PC algorithm to a selected set of variables with an alpha value of 0.05 resulted in physically meaningful graphs. However, due to concerns raised by the reviewers regarding the variable selection process, we removed this step and instead applied the PC algorithm to a consistent set of 22 variables for each runoff signature in the revised manuscript. We acknowledge that the alpha value significantly influences the algorithm's results; however, the relatively small sample size (around 670 data points) may also contribute to graph sparsity or potential underfitting (Zuk et al., 2012).

Therefore, to address the loss of information on potential v-structures and account for the relatively small sample size, we applied the PC algorithm to all 22 variables without performing variable selection, using a significance threshold of 0.2 to allow for a more inclusive initial edge selection. In this setting, the edge assumptions, which are explained in the first comments, avoid appearances of spurious links. To assess the stability of the discovered edges, we performed 1,000 bootstrap resamples of the data and applied the PC algorithm to each resample, using a significance threshold of 0.05 for conditional independence tests. We then measured the strength of each edge based on its frequency across the bootstrap iterations. The resulting edge strength estimates, which represent the proportion of bootstrap samples in which each edge appears, were then mapped onto the initial CPDAG obtained from the PC algorithm. This approach enabled us to evaluate the stability of the inferred causal relationships. This method is inspired by the work of Petersen et al. (2021), which you mentioned in your sixth comment. We addressed these points in Section 3.2 and reflected them in the Results and Discussion sections accordingly.

4. **Furthermore, the authors are mainly interested in discovering the parent set of the outcome Y. Why are you not considering methods which are designed for this task? I mean Invariant Causal Prediction (ICP). I would spend a bit more time on the limits of causal discovery and non-linear methods. I would recommend reading: 1) Model-Based Causal Feature Selection for General Response Types (2024); 2) Invariant Causal Prediction for Nonlinear Models (2018); 3) Causal inference by using invariant prediction: identification and confidence intervals (2016).**

Thank you for your suggestion. While the main focus of this study is indeed on identifying the

causal parents of the runoff signatures, one of the broader objectives is to construct interpretable DAGs that reflect the underlying hydrological processes. These graphs aim to provide a more comprehensive understanding of the system beyond the immediate causal predictors of runoff signatures. We also examined model performance across clusters defined by climate, soil, geology, topography, and vegetation categories. The availability of DAGs allowed us to explore the causal relationships among variables within each category.

We appreciate the recommendation to consider Invariant Causal Prediction (ICP). We have explored the use of ICP as part of our preliminary analysis; however, we found that applying it meaningfully in this context, especially given the non-linear dependencies in hydrological data, requires further investigation and careful adaptation, which is beyond the scope of the current study. Nonetheless, we agree that ICP is a promising approach, and we included a discussion of its potential, highlighting it as an avenue for future research. We discussed this method as a potential approach for future work in the context of our study in the Discussion section.

5. **Furthermore, you are using GAMs without specifying which GAMs you are using. Which link function do you use? Which family of distribution is assumed for the outcome y? Generalized linear models are then useful if your outcome is discrete (Poisson, ...), binary (logistic, c-log-log, ...), or continuous but only positive valued (exponential, gamma, ...). I see why you are using the additive components (splines), but I miss some more information about the model specification. Without this information, reproducibility is not possible.**

Thank you for your comment. We agree that more details on the model specification are important for clarity and reproducibility. In our study, we used Generalized Additive Models (GAMs) implemented via the *mgcv* package in R. Specifically, we used cubic regression splines ($bs = "cr"$) for the smooth terms. The outcome variable is continuous, and we used the default identity link function with a Gaussian error distribution ($family = gaussian()$). The GAMs were fitted using Restricted Maximum Likelihood ($REML$) to estimate the smoothing parameters. We explained the model's specifications in Section 3.3.2.

6. **A further point to note is that while the PC algorithm basically only uses (partial) correlation, testing linear dependency, the subsequent use of additive models assumes a potentially non-linear association between X and the mean of Y. This is not necessarily a flaw in your approach, but it is crucial to be aware that while the PC algorithm relies on (partial) correlations, which inherently assess linear relationships (keeping in mind the fundamental connection between linear regression and correlation), the application of GAMs (the additive part of it) implies that a non-linear relationship between the predictors and the mean of y is assumed. What you could try is something like using regression modeling with cubic spline as a heuristic test of conditional independence. See for an data example: "Data-Driven Model Building for Life-Course Epidemiology (2021)".**

Thank you for this helpful point. We fully agree that the PC algorithm, in its standard form, relies on (partial) correlations to test for conditional independence, which inherently assumes linear relationships among variables. However, given the known nonlinearity of hydrological processes, we tried to address this limitation in two ways.

First, for the conditional independence tests during causal discovery, we used a non-parametric approach based on mutual information with the James-Stein shrinkage estimator (Hausser and Strimmer, 2009), as implemented in the *bnlearn* package. This method does not assume linearity and is better suited than partial correlation for capturing the complex dependencies commonly found in environmental and hydrological systems.

Second, following your suggestion, we used cubic spline regression to further explore potential non-linear associations and assess the robustness of the inferred edges. This served as a heuristic test to evaluate the significance of connections between adjacent nodes in the learned DAG, aligning with your suggestion. Using these spline-based models, we assessed the significance of relationships between adjacent variables in the DAGs by reporting the p-values from likelihood ratio tests for each edge.

We clarified these methodological choices in Section 3.2 of the manuscript, emphasizing that our approach complements the PC algorithm by addressing its linearity assumption both during and after the structure learning phase.

**References**

Hausser, J. and Strimmer, K.: Entropy inference and the James-Stein estimator, with application to nonlinear gene association networks., Journal of Machine Learning Research, 10, 2009.

Petersen, A. H., Osler, M., and Ekstrom, C. T.: Data-Driven Model Building for Life-Course Epidemiology, AMERICAN JOURNAL OF EPIDEMIOLOGY, 190, 1898–1907, https://doi.org/10.1093/aje/kwab087, 2021.

Zuk, O., Margel, S., and Domany, E.: On the number of samples needed to learn the correct structure of a Bayesian network, arXiv preprint arXiv:1206.6862, 2012.

**List of Changes in the Manuscripts**

The list of changes in the manuscript is as follows:

Table 1: Changes in the manuscript.

| Section | Changes |
|---|---|
| Section 1 (Introduction) | Add a short introduction on the PC algorithm, and remove the section about the variable selection. |
| Section 2 (Data) | add more explanation about how the data is used for clustering, causal discovery, and prediction. Modify Table 1 to clarify the data used in the study. |
| Section 3 (Methods) | Add explanation about causal discovery, including causal discovery with the PC, background knowledge, and PC implementation. Add more information about GAM specification. |
| Section 4 (Results) | Update the whole Section 4.2, 4.3, and 4.4 according to the new results. |
| Section 5 (Discussion) | Update this section according to the new results. |
| Section 6 (Conclusion) | Update this section according to the new results. |
| Appendix A | Update the tables in this section according to the new results. |
| Supplementary Materials | Update sections 2 and 3, according to the new results. |